# "ScatSpotter" — A Dog Poop Detection Dataset

## Abstract

We introduce a new dataset containing phone images of dog feces, annotated with manually drawn or AI-assisted polygon labels. Its over 9000 "before/after/negative" full resolution images contain 6000 polygon annotations. The collection and annotation of images started in late 2020. This paper focuses on two checkpoints from 2025-04-20 and 2024-07-03. We train VIT and MaskRCNN baseline models to explore the difficulty of the dataset. The best model achieves a pixelwise average precision of 0.858 on a 691-image validation set and 0.810 on a small independently captured 121-image contributor test set. Dataset snapshots are available through four different distribution methods: two centralized (Girder and HuggingFace) and two decentralized (IPFS and BitTorrent). We study of the trade-offs between distribution methods and discuss the feasibility of each with respect to reliably sharing open scientific data. The code for experiments is hosted on GitHub. The data license is CC-BY 4.0. Model weights are available with the dataset. Experiment hardware, time, energy, and emissions are quantified.

## 1 Introduction

Applications for a computer vision system capable of detecting and localizing poop in images are numerous. These include automated waste disposal to keep parks and backyards clean, tools for monitoring wildlife populations via droppings, and a warning system in smart-glasses to prevent people from stepping in poop. Our primary motivating use case is a phone application that assists dog owners in locating their dog's poop in a leafy park for easier cleanup. Many of these applications can be realized with modern object detection and segmentation methods [48, 50, 55] combined with a large labeled dataset.

In addition to enabling several applications, poop detection is an interesting benchmark problem. It is relatively simple, with a narrow focus on a single class, making it suitable for exploring the capabilities of object detection models that target a single labeled class. However, the task includes non-trivial challenges such as resolution issues (e.g., camera quality, distance), camouflaging distractors (e.g., leaves, pine cones, sticks, dirt, and mud), occlusion (e.g., bushes, overgrown grass), and variation in appearance (e.g., old vs. new, healthy vs. sick). An example of a challenging case is shown in Figure 1a. Investigation into cases where this problem is difficult may provide insight into how to better train object detection and segmentation networks.

Towards these ends we introduce a new dataset which, in formal settings, we call "ScatSpotter". Poops are annotated with polygons making the dataset suitable for training detection and segmentation models. In order to assist with annotation and add variation, we collect images using a "before/after/negative" (BAN) protocol as shown in Figure 1b.

From this data, we train a segmentation model to classify which pixels in an image contain poop and which do not. Our models show strong performance, but there are notable failure cases indicating this problem is difficult even for modern computer vision algorithms.

Submitted to 39th Conference on Neural Information Processing Systems (NeurIPS 2025). Do not distribute.

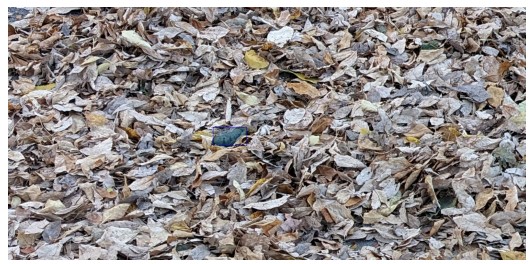 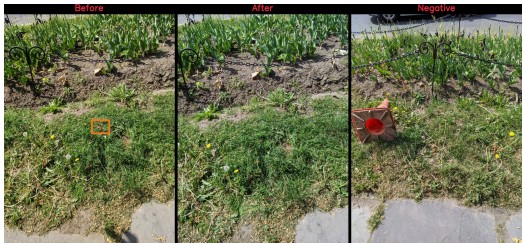

(a) A zoomed in example of an annotated object in a challenging condition: a scene cluttered with leaves. The similarity between the leaves and the poop causes a camouflage effect that can make detecting it difficult. The poop is highlighted in blue.

(b) The "before/after/negative" protocol. The orange box highlights the location of the poop in the "before" image. In the "after" image, it is the same scene but the poop has been removed. The "negative" image is a nearby similar scene, potentially with a distractor. Note that the object is small relative to the image size.

Figure 1: (a) A challenging annotation case due to camouflage. (b) The BAN protocol.

Table 1: Related datasets. Columns list dataset name, number of categories, images, and annotations. Image W × H gives median image dimensions; Ann Area$^{0.5}$ is the median square root of annotation area (pixels); Size is disk requirements in GB; Annot Type is the labeling method. Figure 2 shows the distribution of annotation shapes, sizes, and locations.

| Name | #Cats | #Images | #Annots | Image W × H | Annot Area$^{0.5}$ | Disk Size | Annot Type |
|---|---|---|---|---|---|---|---|
| ImageNet[47] | 1,000 | 594,546 | 695,776 | 500 × 374 | 239 | 166GB | box |
| MSCOCO[33] | 80 | 123,287 | 896,782 | 428 × 640 | 57 | 50GB | polygon |
| CityScapes[12] | 40 | 5,000 | 287,465 | 2,048 × 1,024 | 50 | 78GB | polygon |
| ZeroWaste [3] | 4 | 4,503 | 26,766 | 1,920 × 1,080 | 200 | 10GB | polygon |
| TrashCanV1[25] | 22 | 7,212 | 12,128 | 480 × 270 | 54 | 0.61GB | polygon |
| UAVVaste[29] | 1 | 772 | 3,718 | 3,840 × 2,160 | 55 | 2.9GB | polygon |
| SpotGarbage[40] | 1 | 2,512 | 337 | 754 × 754 | 355 | 1.5GB | category |
| TACO[45] | 60 | 1,500 | 4,784 | 2,448 × 3,264 | 119 | 17GB | polygon |
| MSHIT[38] | 2 | 769 | 2,348 | 960 × 540 | 99 | 4GB | box |
| Ours | 1 | 9,296 | 6,594 | 4,032 × 3,024 | 87 | 60GB | polygon |

To enable others to build on our results, it is essential that the dataset is accessible and hosted reliably. Centralized methods are a typical choice, offering high speeds, but they can be costly for individuals, often requiring institutional support or paid hosting services. They are also prone to outages and lack built-in data validation. In contrast, decentralized methods allow volunteers to host data and offers built-in validation of data integrity. This motivates us to compare and contrast the decentralized BitTorrent [8], and IPFS [4] protocols as mechanisms for distributing datasets.

Our contributions are: 1) A challenging new **open dataset** of images with polygon annotations. 2) A set of trained **baseline models**. 3) A **comparison of dataset distribution** methods.

## 2 Related Work

To the best of our knowledge, our dataset is currently the largest publicly available collection of annotated dog poop images, but it is not the first. A dataset of 100 dog poop images was collected and used to train a FasterRCNN model [42] but this dataset and model are not publicly available. The company iRobot has a dataset of annotated indoor poop images used to train Roomba j7+ to avoid collisions [21], but as far as we are aware, this is not available. In terms of available poop detection datasets we are only aware of MSHIT [38] which is much smaller, only contains box annotations, and the objects of interest are plastic toy poops.

Compared to benchmark object localization and segmentation datasets [47, 33, 12] ours is much smaller and focused only on a single category. However, when compared to litter and trash datasets

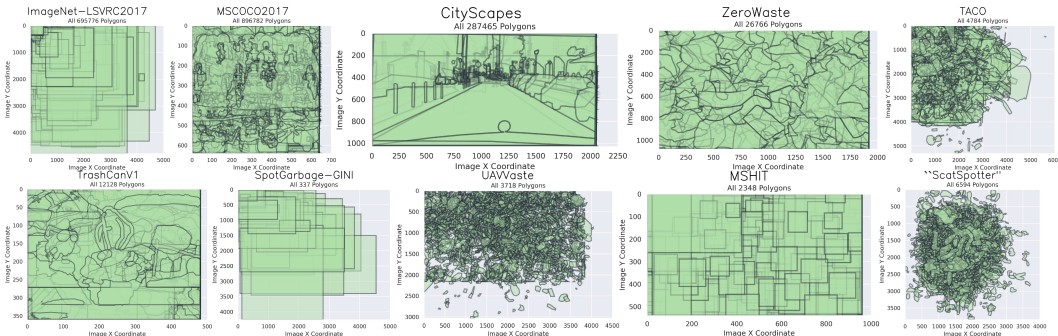

Figure 2: A comparison of all of the annotations for different datasets including ours. All polygon annotations drawn in a single plot with 0.8 opacity to demonstrate the distribution in annotation location, shape, and size with respect to image coordinates.

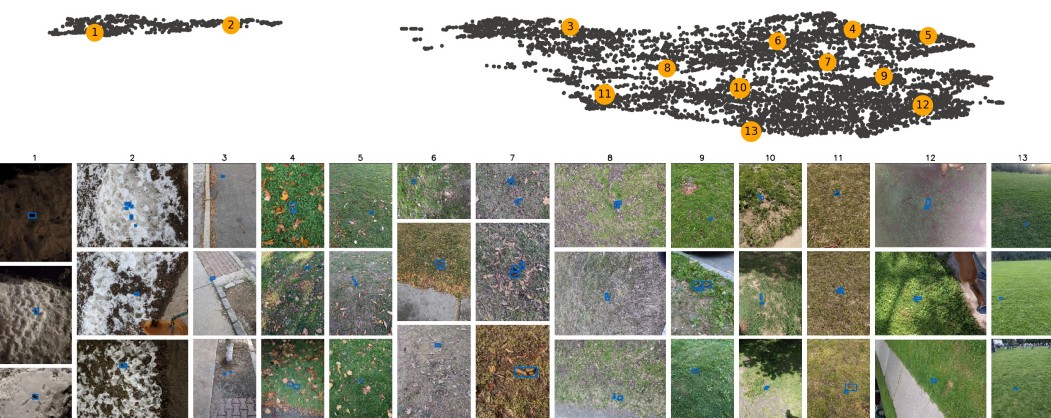

Figure 3: Example images from 2D UMAP clusters [37]. Each point in the top image represents a 2D-projected embedding, with numbered orange dots indicating nearby images in the bottom columns. Blue annotation boxes are shown. A clear separation emerges between snowy (columns 1-2) and non-snowy images (columns 3-13).

[3, 45, 25, 40, 29] ours is among the largest in terms of number of images / annotations, image size, and total dataset size. ZeroWaste [3] uses a "before/after" protocol similar to our BAN protocol. We provide an overview of these related datasets in Table 1. Among all of these, ours stands out for having the highest resolution images and the smallest objects relative to that resolution. For a review of additional waste related datasets, refer to [39].

Section 5 discusses the logistics and tradeoffs between dataset distribution mechanisms with a focus on comparing centralized and decentralized methods. IPFS [4] and BitTorrent [8] are the decentralized mechanisms we evaluate, but others exist such as Secure Scuttlebut [52] and Hypercore [17], which we did not test.

# 3 Dataset

Our first contribution is the creation of a new open dataset which consists of images of dog poop in mostly urban, mostly outdoor environments, from mostly a single city. The data is annotated to support object detection and segmentation tasks. The majority of the images feature fresh poop from three specific medium sized dogs, but there are a significant number of images with poops of unknown age and from unknown dogs.

Despite these biases, the dataset has significant image variations. To provide a gist, we computed UMAP [37] image embeddings based on ResNet50 [22] descriptors display images corresponding with clusters in this embedding in Figure 3.

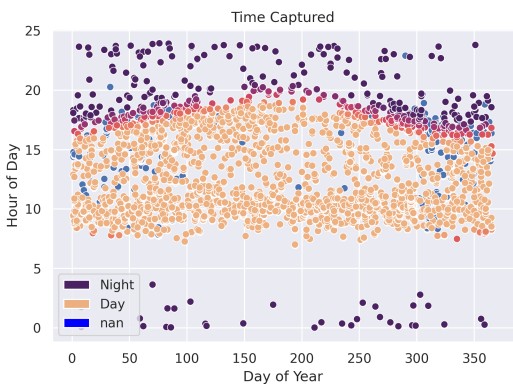
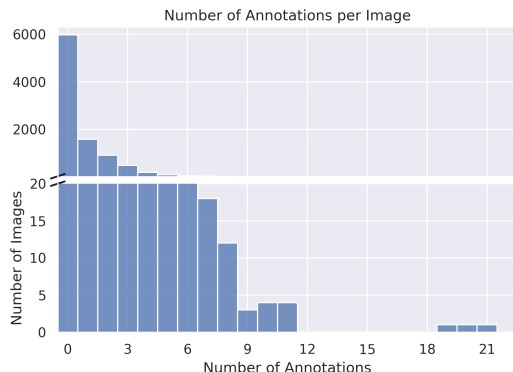

(a) The time-of-year vs time-of-day of each image show lighting and seasonal variation. On the x-axis, 0 is January 1st. On the y-axis, 0 is midnight. Color estimates daylight based on location (if available). Most images are in the day, but many were taken at night with flash or long exposure.

(b) The histogram of annotations per image shows object density variation. Only 35% (3,314) of images contain annotations; 65% (5,982) are known negatives. About half of the negatives were taken immediately after pickup; the rest are from nearby locations with potential lookalikes.

Figure 4: Dataset distributions. (a) Time and daylight scatterplot. (b) Annotation count histogram.

More details about the dataset are available in a standardized datasheet [18] that covers the motivation, composition, collection, preprocessing, uses, distribution, and maintenance. This will be distributed with the data itself, and is provided in supplemental material.

## 3.1 Dataset Collection

A single researcher on dog walks photographed fresh dog poop, mostly their own dogs, but often others. Distance was sometimes varied for diversity. Most images were taken following the "before/after/negative" (BAN) protocol. A BAN triple comprises a "before" shot of the poop, an "after" shot post removal, and a "negative" shot of a nearby lookalike (e.g., pine cones, leaves). We only use them for negative sampling, but they could enable contrastive triplet losses [49].

The majority of images follow the BAN protocol, but there are exceptions. The first six months of data collection only involved the "before/after" part of the protocol. We began collecting the third negative image after a colleague suggested it. In some cases, the researcher failed or was unable to take the second or third image. These exceptions are often programmatically identifiable.

We also received 121 contributor images, mostly outside the BAN protocol. These images are held out and used as our test set. Due to the small size, our main results also include validation scores.

## 3.2 Dataset Annotation

Images were annotated using labelme [27]. Most annotations were initialized using SAM and a point prompt. All AI polygons were manually reviewed. In most cases only small manual adjustments were needed, but there were a significant number of cases where SAM did not work well and fully manual annotations were needed. Regions with shadows seemed to cause SAM the most trouble, but there were other failure cases. Unfortunately, there is no metadata to indicate which polygons were manually created or done using AI. However, the number of vertices may be a reasonable proxy to estimate this, as polygons generated by SAM tend to have higher fidelity boundaries. The boundaries of the annotated polygons are illustrated in Figure 2.

Data collected after 2024-07-03 was annotated with the help of models trained on prior data. Again, all predictions were manually verified or corrected. In these later cases, false positive annotations were labeled (e.g. stick, leaf), but because these categories are not labeled exhaustively, we exclude them from all analysis in this paper.

### 3.3 Dataset Properties and Statistics

The data was captured at a regular rate over 4.3 years, primarily in parks and sidewalks within a small city. Weather conditions varied across snowy, sunny, rainy, and foggy. A visual representation of the distribution of seasons, time-of-day, daylight, and capture rate is provided in Figure 4a.

The dataset images are available in full resolution. Almost all images were taken using the same phone-camera, with a consistent width/height of 4,032 × 3,024 (although some may be rotated based on EXIF data). The images are stored as 8-bit JPEGs with RGB channels, and most include overviews (i.e., image pyramids), allowing for fast loading of downscaled versions.

Due to the BAN protocol, about one-third of the images contain annotations, the rest were taken after the object(s) were removed. Consequently, most images have no annotations. When present, annotations are usually singular, but multiple annotations are common and can be due to: 1) fragmented dropping, 2) dogs pooping together, 3) repeated poops in the same area over time (sometimes hard to distinguish from dirt). The number of annotations per image is illustrated in Figure 4b.

### 3.4 Dataset Splits

Our dataset is split into training, validation, and test sets based on the year and day of image capture and photographer. Only data captured by the authors is used for training and validation. Of these, images from 2021-2023, 2025 and beyond are assigned to the training set. Images from 2020 are used for validation. For data from 2024, we consider the ordinal date $n$ of each image and include it in the validation set if $n \equiv 0 \pmod 3$; otherwise, it is assigned to the training set.

For testing data, we use contributor images to not bias our results based on the way the authors took images. These splits are provided in the COCO JSON format [33] as well as a WebDataset [53] on HuggingFace.

## 4 Baseline Models

As our second contribution, we trained and evaluated models to establish a baseline for future comparisons. Specifically we train three model variants. We trained two MaskRCNN [23] models (specifically the `R_50_FPN_3x` configuration), one starting from pretrained ImageNet weights (MaskRCNN-p), and one starting from scratch (MaskRCNN-s). We also trained a semantic segmentation vision transformer variant (VIT-sseg-s) [20, 13], which was only trained from scratch. Hyperparameters are given in supplemental materials.

For these baseline models, the training data was limited to an older subset taken before 2024-07-03. Our training dataset consists of 5,747 images and is identified by a suffix of `1e73d54f`, which is the prefix of its content hash. The validation set contains 691 images and has a suffix of `99b22ad0`. The test set, consists of the 121 images, has a suffix of `6cb3b6ff`, and includes contributor images up to 2025-04-20. The evaluated models were selected based on their validation scores.

We performed two types of evaluations on the models. "Box" evaluation computes standard COCO object detection metrics [33]. MaskRCNN natively outputs scored bounding boxes, but for the VIT-sseg model, we convert heatmaps into boxes by thresholding the probability maps and converting taking the extend of the resulting polygons as bounding boxes. The score is taken as the average heatmap response under the polygon. Bounding box evaluation has the advantage that small and large annotations contribute equally to the score, but it can also be misleading for datasets where the notion of an object instance can be ambiguous.

To complement the box evaluation, we performed a pixelwise evaluation, which is more sensitive to the details of the segmented masks, but also can be biased towards larger annotations with more pixels. The corresponding truth and predicted pixels were accumulated into a confusion matrix, allowing us to compute standard metrics [44] such as precision, recall, false positive rate, etc. For the VIT-sseg model, computing this score is straightforward, but for MaskRCNN we accumulate per-box heatmaps into a larger full image heatmap, which can then be scored.

Quantitative results for each of these models on box and pixel metrics are shown in Table 2. Because the independent test set is only 121 images, we also present results on the larger validation dataset. Corresponding qualitative test results are illustrated in Figure 5 and validation results in Figure 6.

Table 2: Results for MaskRCNN and VIT models (suffix -p: pretrained, -s: scratch) on test and validation sets. Evaluated with box and pixel metrics — AP (ppv-tpr area) [44] and AUC (tpr-fpr area) — computed via scikit-learn [43]. Pretrained models outperform. Note: VIT-sseg was tuned more; MaskRCNN may yield better results with similar effort.

| Dataset split: | | Test (n=121) | | | | Validation (n=691) | | | |
| Evaluation type: | | Box | Box | Pixel | Pixel | Box | Box | Pixel | Pixel |
| Model type | # Params | AP | AUC | AP | AUC | AP | AUC | AP | AUC |
| --- | --- | --- | --- | --- | --- | --- | --- | --- | --- |
| MaskRCNN-p | 43.9e6 | **0.613** | **0.697** | **0.810** | 0.849 | **0.612** | **0.721** | **0.858** | 0.905 |
| MaskRCNN-s | 43.9e6 | 0.253 | 0.464 | 0.384 | 0.798 | 0.255 | 0.576 | 0.434 | 0.891 |
| VIT-s | 25.5e6 | 0.422 | 0.426 | 0.473 | **0.902** | 0.476 | 0.532 | 0.780 | **0.994** |

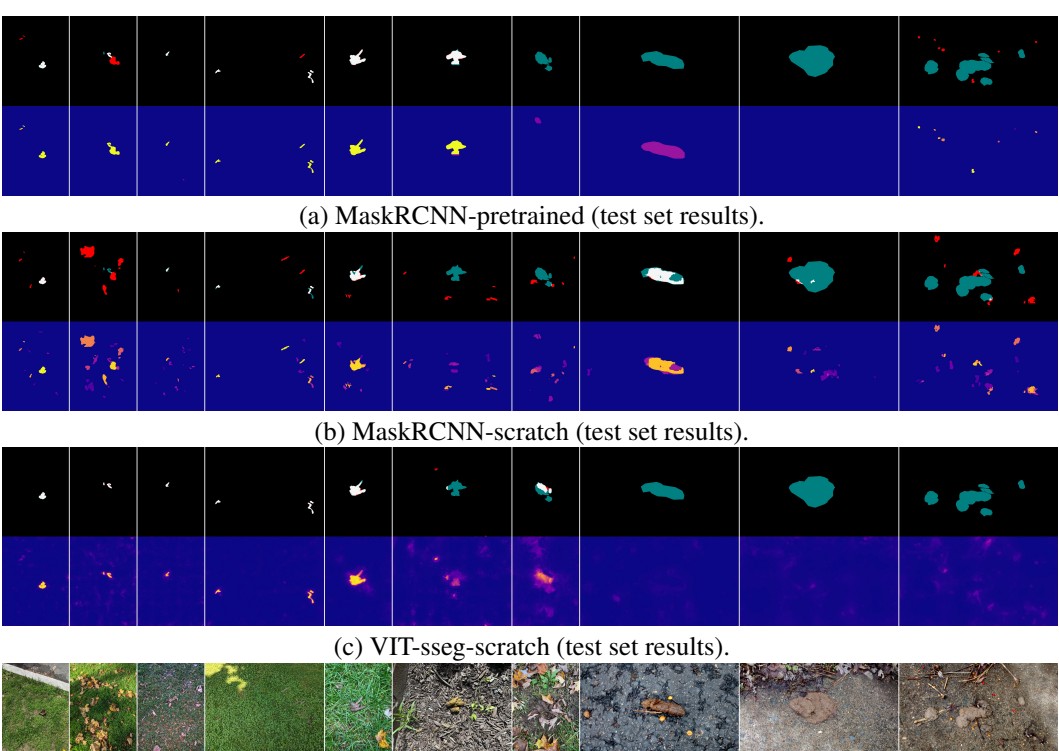

(a) MaskRCNN-pretrained (test set results).

(b) MaskRCNN-scratch (test set results).

(c) VIT-sseg-scratch (test set results).

(d) Input images from the test set.

Figure 5: Qualitative results from the top model on the validation set, applied to test images. The first three subfigures (a, b, c) display a binarized classification map (true positives in white, false positives in red, false negatives in teal, true negatives in black) and the predicted heatmap (before binarization). Subfigure (d) shows the input image. The heatmap binarization threshold was 0.5. Failures occur with close-up or deteriorated objects, and camouflage.

All models were trained on a single machine with an Intel Core i9-11900K CPU and an NVIDIA GeForce RTX 3090 GPU. A key limitation of these results is the imbalance between model types, with 42 out of 44 trained models being VIT-ssegs and only two MaskRCNN models, each taking approximately 8 hours to train. Future work could further optimize MaskRCNN models to improve comparability. More details on the VIT-sseg experiments can be found in the supplemental materials.

**Environmental Impact** The total time spent on prediction and evaluation across all experiments was 15.6 days, with prediction consuming 109.63 kWh of energy and causing an estimated emissions of 23.0 $CO_2$kg as measured by CodeCarbon [30]. We estimated train-time resource usage during training using indirect methods, assuming a constant power draw of 345W from the RTX 3090 GPU. Energy consumption was approximated accordingly, while emissions were calculated using a conversion ratio of 0.21 $\frac{\text{kgCO}_2}{\text{kWh}}$ derived from our prediction time measurements. Based on file timestamps, we

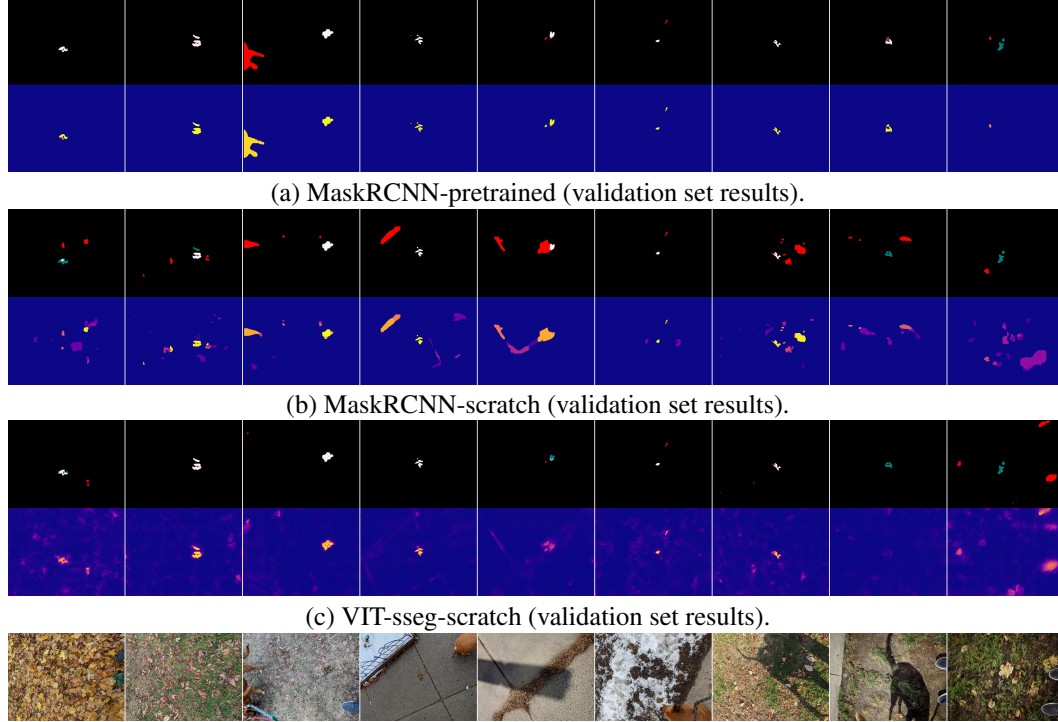

(a) MaskRCNN-pretrained (validation set results).

(b) MaskRCNN-scratch (validation set results).

(c) VIT-sseg-scratch (validation set results).

(d) Inputs from the validation set.

Figure 6: Qualitative results of the top model on unseen validation images (see Figure 5 for visualization details). Although never trained on these data, the model's was able to detect camouflaged cases on the left but missed some on the right, indicating generalizability but also room for improvement.

estimated that running 44 different training runs took approximately 159.66 days, resulting in an estimated energy usage and emissions of 1321.99 kWh and 277.612 $CO_2$ kg, respectively. For context, at $\frac{\$0.16}{\text{kWh}}$ and $\frac{\$25.00}{1000CO_2\text{kg}}$, the cost of training and evaluating was $229.06.

# 5    Open Data Distribution

Empirical evidence suggests that a substantial proportion of scientific studies have low reproducibility rates, which has raised concerns across various disciplines [2]. Ideally, scientific research should be independently reproducible. Despite higher success rates in computer science (up to 60%) compared to other fields, there is still room for improvement [46, 11, 14]. Addressing this issue requires not just better experimental documentation but also more reliable and accessible data distribution methods. Specifically, this involves robustly codifying data download and preparation processes.

Centralized data distribution methods allow for codified data access by storing URLs that point to datasets within the code, offering fast and direct access. However, this approach lacks robustness. It can fail if the provider goes offline, changes the URL, or stops hosting the data. Additionally, cloud storage can be expensive, and users must trust that the provider delivers the correct data — a risk that can be mitigated by using checksums to verify data integrity.

In contrast, decentralized methods allow users to access data in the same way, even if the organization hosting the data changes. By leveraging content-addressable storage, where the dataset checksum acts as both the key to locate and validate the data, these methods ensure data integrity and nearly eliminate the risk of dead URLs, provided that at least one peer retains the data. While decentralized systems face challenges such as longer connection times, increased network overhead, and the need for a robust peer network, their ability to ensure data access via a static address motivates our investigation

Specifically, we focus on two prominent candidates: BitTorrent and IPFS. BitTorrent [8, 9] is a well known sharing protocol that originally relied on centralized trackers and databases of torrent files

to connect peers. While trackers and torrent files are still prominent, torrents can be published to a distributed hash table (DHT) using the Kademlia algorithm [36]. This makes it an strong candidate for a decentralized distribution mechanism. On the other hand, IPFS (InterPlanetary File System) [4, 6] is a newer tool directly build directly on a DHT. IPFS has been likened to "a single BitTorrent swarm, exchanging objects within one Git repository". Both IPFS and BitTorrent are content addressable at the dataset level, which makes them both appropriate for our use case where we seek a static address that can be used to robustly access data.

It is worth noting that git-based [7] systems like HuggingFace [32] with large file storage do gain some decentralized properties via multiple remotes, but not content identifiers.

For practitioners, key concerns are how quickly and reliably data can be accessed. By comparing decentralized and centralized mechanisms access times for our dataset, we aim to make explicit the tradeoffs between the methods and inform decisions on adopting an approach.

## 5.1 Dataset Transfer Experiment

Our third contribution is an experiment that studies transfer rates of decentralized and centralized data distribution methods. For centralized distribution, we use a self-hosted instance of Girder [41] and the HuggingFace datasets [32] platform. For decentralized clients, we use Transmission [31] (BitTorrent) and Kubo [26] (IPFS). As a baseline, we also measure direct transfers using Rsync [54].

For data transfer experiments, we use the 2024-07-03 version of the dataset. This is content-addressed with the IPFS CID (content identifier): `bafybeiedwp2zvmdyb2c2axrcl455xfbv2mgdbhgkc3dil e4dftiimwth2y` The torrent has a magnet URL of: `magnet:?xt=urn:btih:ee8d2c87a39ea9bf e48bef7eb4ca12eb68852c49`, and is tracked on Academic Torrents [10].

To assess the effectiveness of each mechanism we programmatically download our 42GB dataset and measure the time required to complete the transfer. Each experiment was run five times, machines we controlled were separated by $\sim 30$ kilometers with an average ping time of 48.48 ms. For each test, we log transfer start and end times along with notes and code (provided in supplemental materials).

While our measurements provide a reasonable estimate of for access time for each mechanism, there are notable limitations in our methodology. First, different machines and networks have different upload and download speeds, and network congestion is variable. For decentralized methods, we lack an automated mechanism separate peer-connection time and actual download time. Additionally, Girder and HuggingFace required data to be packed into compressed archives, improving transfer efficiency due to fewer file boundaries. In decentralized cases, we provide granular access to each file in the dataset, which avoids an extra unpacking step and enables sharing of the same file between different versions of the datasets and simpler updates, but decreases transfer efficiency. Due to this, we provide both a compressed and uncompressed rsync baseline. Another confounding factor is that with decentralized mechanisms the number of seeders is not controlled for. Subsets of the data have been hosted on IPFS for years, and portions of the dataset may be provided by unknown members of the network. For BitTorrent, our initial transfers only had one seeder, but during our tests other nodes accessed and started to provide the data.

Despite significant testing limitations, our measurements quantify the expected data-access time penalty to gain the advantages of decentralized mechanisms. With these limitations acknowledged, we present the transfer times statistics in Table 3. Alongside these measurements, several observations are worth noting. Transferring files using IPFS had significantly delayed peer discovery times, and we were only able to connect two machines after manually informing them of each other's peer ID. For BitTorrent, were unable to use the mainline DHT and fell back to using trackers. We believe these peer discovery issues are because the dataset has a small number of seeders. To test this, we downloaded other established datasets via IPFS and BitTorrent and found that the peer discovery time was almost immediate, suggesting that this becomes less of an issue as a dataset is shared. However, the inability to quickly find a nearby peer is a major issue for initial or private dataset development.

The HuggingFace results stand out, as they are faster than rsync. We believe this is due to an optimized client and content delivery networks, utilizing CAKE [24] to minimize buffer bloat [19]. However, this speed relies on costly centralized infrastructure. The expected speed from a more modest centralized service is $\sim 20\times$ slower.

Table 3: Transfer times (in hours) for our 42GB dataset: trials (n), mean ($\mu$), std ($\sigma$). Each experiment was run 5 times. Uncompressed transfers provide granular access to individual files, while compressed transfers are faster.

| Method | Compressed | $\mu$ | $\sigma$ | Min | Max |
|---|---|---|---|---|---|
| BitTorrent | No | 8.36h | 5.16h | 2.21h | 14.39h |
| IPFS | No | 10.68h | 9.54h | 1.80h | 24.62h |
| Rsync | No | 4.84h | 1.39h | 3.10h | 6.10h |
| Girder | Yes | 2.85h | 2.31h | 1.05h | 6.24h |
| HuggingFace | Yes | **0.14h** | 0.03h | 0.11h | 0.18h |
| Rsync | Yes | 1.10h | 0.03h | 1.07h | 1.13h |

There is an additional $\sim 4\times$ slowdown between compressed and uncompressed rsync baselines, which needs to be considered when comparing decentralized results. The minimum time column shows that decentralized methods method can be competitive with rsync, but on average decentralized mechanisms are significantly slower and can be stifled by long peer-discovery times.

# 6  Conclusion

We have introduced the largest open dataset of high resolution images with polygon segmentations of dog poop. The dataset contains several challenges including amorphous objects, multi-season variation, difficult distractors, daytime / nighttime variation. We have described the dataset collection and annotation process and reported statistics on the dataset.

We provided a recommended train/validation/test split of the dataset, and trained baseline segmentation models that perform well, but could likely be improved. In addition to providing quantitative and qualitative results of the models, we also estimate the resources required to perform these training, prediction, and evaluation experiments.

We have published our data and models under a permissive license, and made them available through both centralized (Girder and HuggingFace) and decentralized (BitTorrent and IPFS) mechanisms. Decentralized methods have robustness properties, but suffer from significant network transfer overhead. HuggingFace has exceptionally fast transfer speeds, and due to its usage of git-lfs has some decentralized properties, but lacks content identifiers. Combining IPFS with a content distribution network may be a path to a best-of-both-worlds system.

Limitations of our work include: 1) geographic concentration of the dataset, 2) the small size of the independent test set, 3) limited exploration of the better-performing model variant, and 4) uncontrolled network conditions during distribution experiments. Future work could address these by expanding dataset diversity, training a broader range of models, and improving decentralized hosting strategies.

Our dataset enables applications such as mobile apps for detecting feces, urban cleanliness monitoring, and augmented reality collision warnings. We believe negative impacts are limited and expect respectful use of the dataset. We envision exciting possibilities for the BAN protocol in computer vision research. We hope our work will inspire others to consider decentralized content addressable data sharing, fostering open collaboration and reproducible experiments. Furthermore, we encourage the community to track experimental resource usage to better understand and offset our experiments' small, but real environmental impact. Moreover, we aspire for our dataset to enable the creation of poop-aware applications. Ultimately, our goal is for this research to contribute meaningfully to the advancement of computer vision and have a positive impact on society.

# 7  Acknowledgements

We would would like to thank all of the dogs that produced subject matter for the dataset, all of the contributors for helping to construct a challenging test set, and [redacted for peer review] for several suggestions including taking the third negative picture. This work is dedicated to [redacted for peer review], a very weird and very good girl.

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

## A  Expanded Dataset Information

In Section 3 we provided an overview of several dataset statistics. In this appendix we expand on that with additional plots. The distribution of image pixel intensities is illustrated in Figure 7. The distribution of images collected over time is shown in Figure 8. The distribution of annotation location is shown in Figure 9 and sizes is shown in Figure 10 and Figure 11.

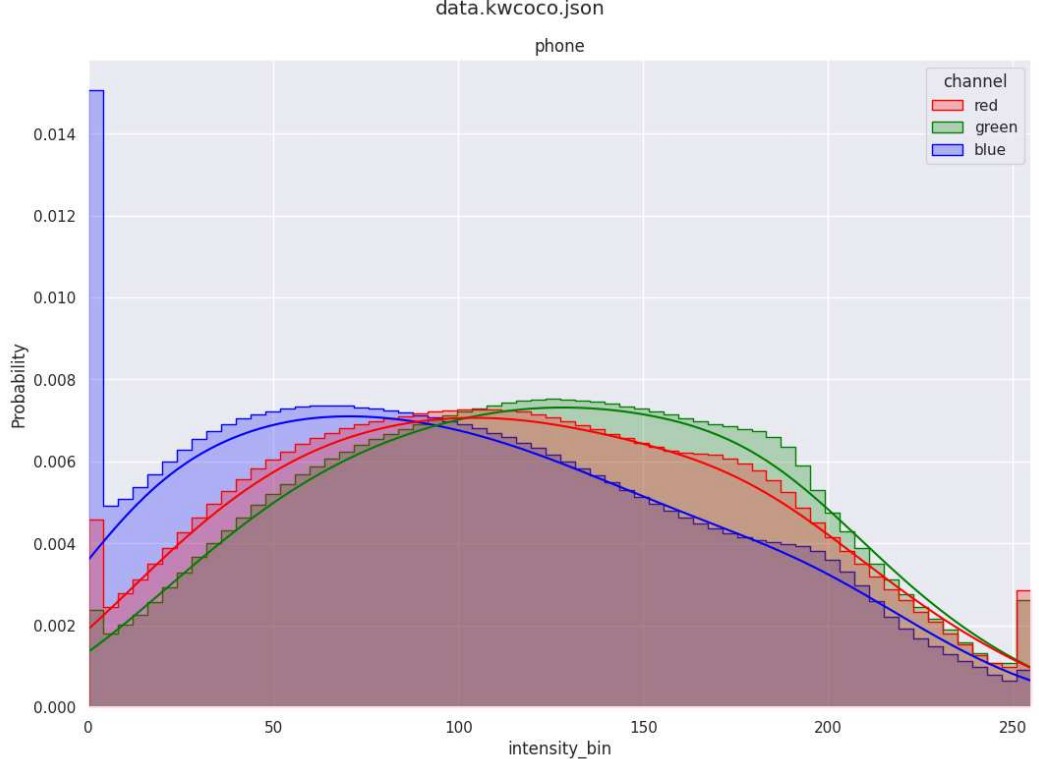

Figure 7: The "spectra" or histogram of the pixel intensities in the dataset. The dataset RGB mean/std is $[117, 124, 100]$, $[61, 59, 63]$. This was run on the older 2024-07-03 snapshot.

## B  Expanded Dataset Comparison

In Section 2 we compared to related work. Here we expand on this by comparing our analysis plots. Every dataset is converted into the COCO format and visualized using the same logic. Figure 2 visualizes the annotations of all datasets. We make similar visualizations for other comparable dataset metrics. Figure 12 shows the number of annotations per image. Figure 13 shows of image sizes in each dataset. Figure 14 shows the distribution of width and heights of oriented bounding boxes fit to annotation polygons. Figure 15 shows the area of each polygon versus the number of vertices (which could be used to estimate the likelihood a polygon was generated by AI for our dataset). Figure 16 shows the distribution of centroid positions (relative to the image size).

## C  VIT-sseg Models

This section provides more details about the training of VIT-sseg models.

To train VIT-sseg models we use the training, prediction, and evaluation system presented in [20, 13], which utilizes polygon annotations to train a pixelwise binary segmentation model.

In all experiments, we use half-resolution images, which means most images have an effective width × height of 2,016 × 1,512. We employ a spatial window size of 416 × 416 for network inputs, which

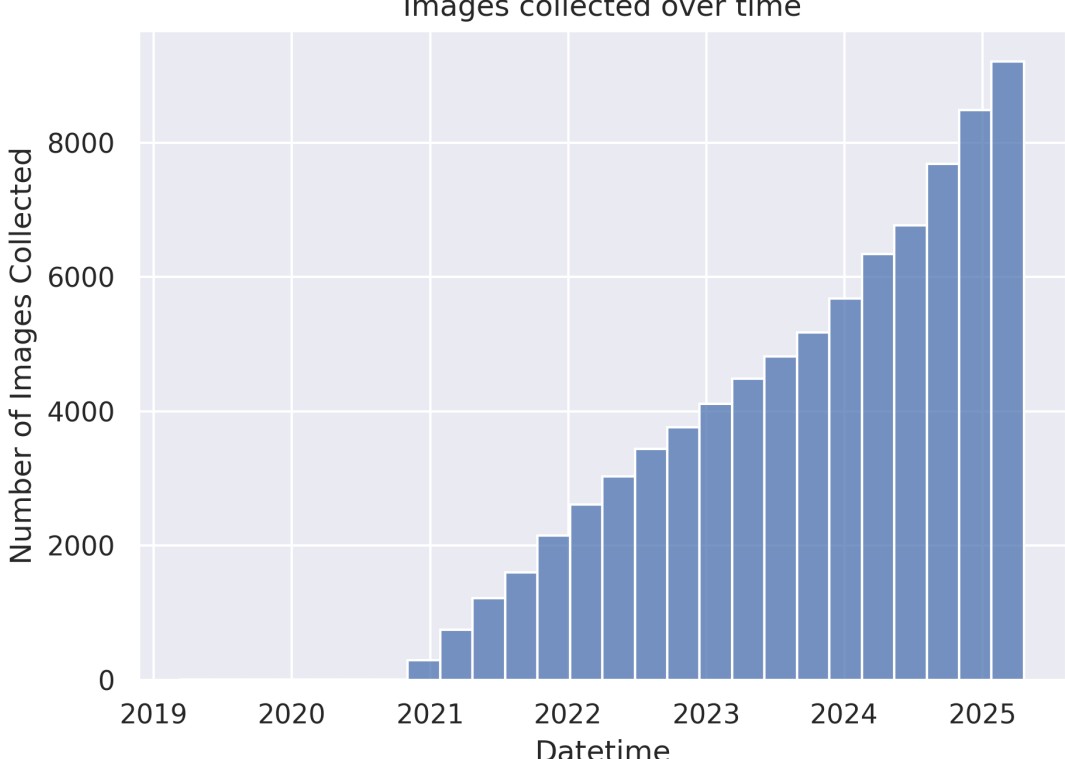

Figure 8: The number of images collected over time.

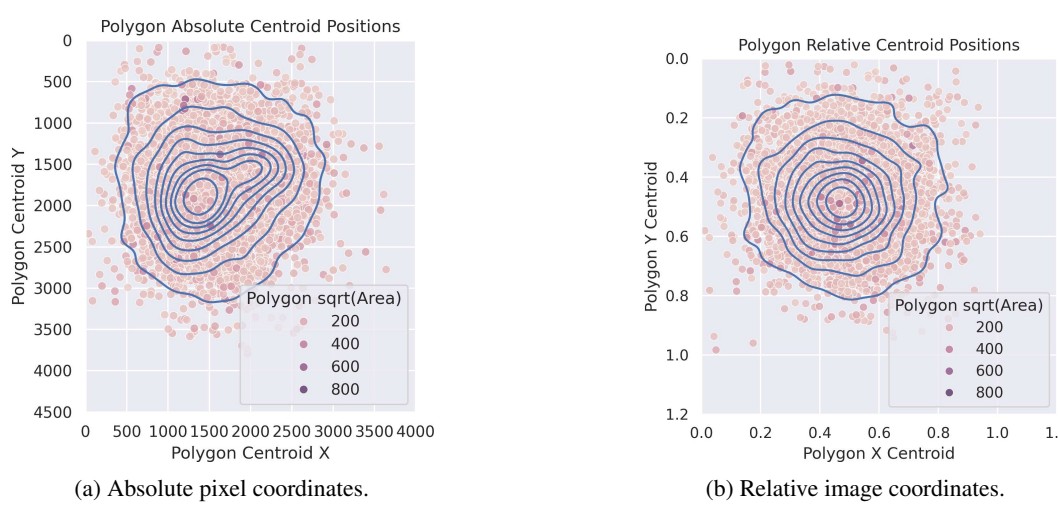

(a) Absolute pixel coordinates.

(b) Relative image coordinates.

Figure 9: The distribution of annotation centroids in terms of (a) absolute image coordinates and (b) relative image coordinates. The absolute centroid distribution is bimodal because some images are taken in landscape mode and other in portrait mode.

means that multiple windows are needed to predict on entire images. During prediction, we apply a window overlap of 0.3 with feathered stitching to prevent boundary artifacts.

To address the class imbalance in our dataset (where positives are patches containing annotations and negatives contain no annotations), we adopt a balanced sampling strategy. Each "epoch" consists of randomly sampling 32,768 patches from the dataset with replacement, ensuring roughly equal numbers of positive and negative samples. We train each network for 163,840 gradient steps. For data augmentation we use random crops and flips.

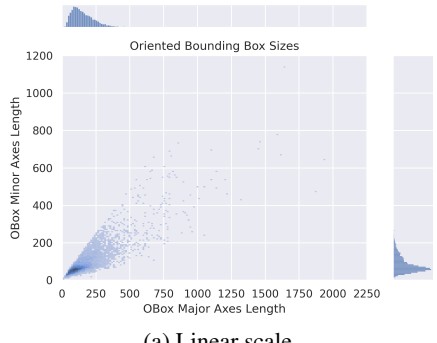 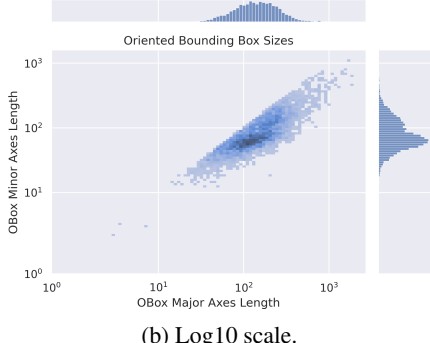

(a) Linear scale.    (b) Log10 scale.

Figure 10: The distribution of annotation sizes as measured by an oriented bounding box fit to each polygon. (a) shows this plot on a linear scale and (b) show this plot on a log scale.

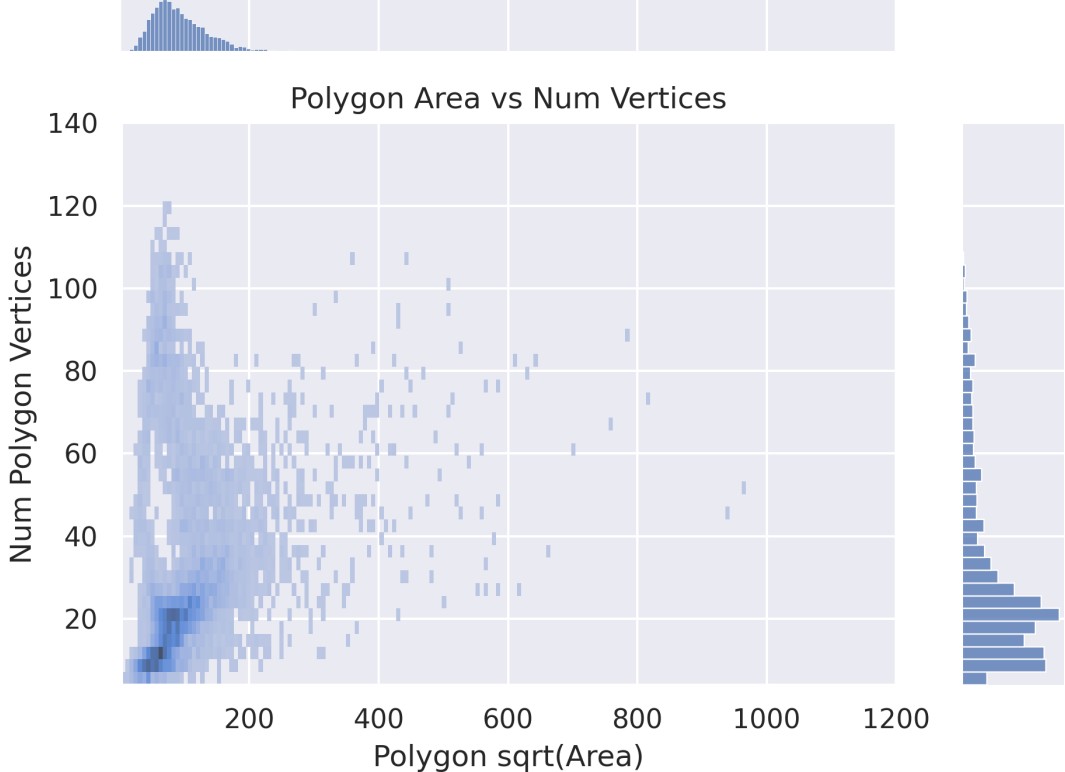

Figure 11: The distribution of polygon areas versus the number of vertices in the polygon boundary. The SAM model tends to produce polygons with a higher number of vertices than manually drawn ones. For smaller polygons there are two peaks in the number of vertices histograms likely corresponding to pure-manual versus AI-assisted annotations.

Our baseline architecture is a variant [5, 20] of a vision-transformer [16]. The model is a 12-layer encoder backbone with 384 channels and 8 attention heads that feeds into a 4-layer MLP segmentation head. It has 25,543,369 parameters and a size of 114.19 MB on disk. At predict time it uses 1.96GB of GPU RAM.

We compute loss pixelwise using Focal Loss [34] with a small downweighting of pixels towards the edge of the window. Our optimizer is AdamW [35], and we experiment with varying learning rate, weight decay, and perturb-scale (implementing the shrink perturb trick [1, 15]). We employ a OneCycle learning rate scheduler [51] with a cosine annealing strategy and starting fraction of 0.3. Our effective batch size is 24 with a real batch size of 2 and 12 accumulate gradient steps. This setup consumes approximately 20 GB of GPU RAM during training.

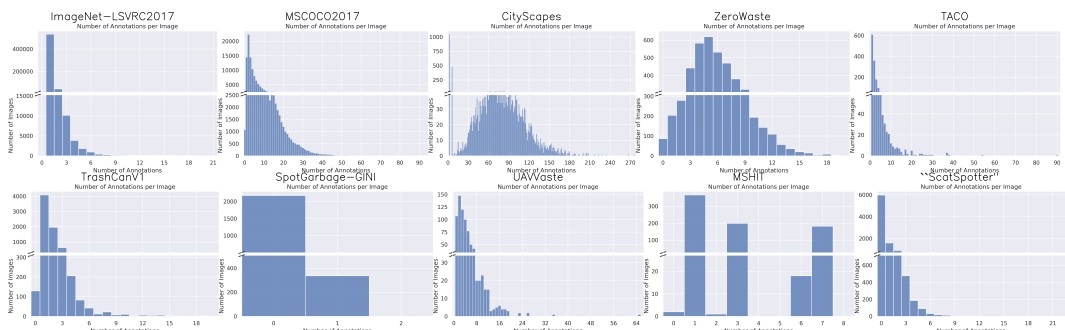

Figure 12: Number of annotations per image in each dataset.

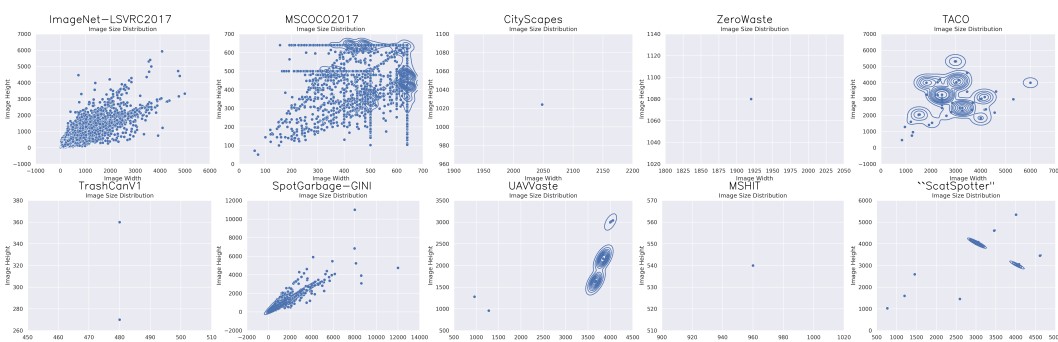

Figure 13: Image size distributions of each dataset. Ours has two primary width/heights.

## C.1 VIT-sseg Model Experiments

To establish a baseline, we evaluated 35 training runs where we varied input resolutions, window sizes, model depth, and other parameters. Although this initial search was somewhat ad-hoc, it provided insights into the optimal configuration for our model. Building on the best hyperparameters from this search, we performed a sweep over 7 combinations of learning rate, weight decay, and perturb scale (i.e., shrink and perturb [1, 15]). Scripts used to reproduce these experiments, as well as a log of the ad-hoc experiments, are available in the code repository. Additionally, trained models are packaged and distributed with information about their training configuration.

Note: the test dataset used in this appendix section is an older 30 image version with suffix d8988f8c, which is a subset of the more recent 121 image test set used in the main paper.

For each of the 7 hyperparameter combinations, we trained the model for 163,840 optimizer steps using a batch size of 24. We defined an "epoch" as 1,365 steps, at which point we saved a checkpoint, evaluated validation loss, and adjusted learning rates. To conserve disk space, we retained only the top 5 lowest-validation-loss checkpoints (although training crashes and restarts sometimes resulted in additional checkpoints, which are included in our evaluation).

Using the top-checkpoints, we predicted heatmaps for each image in the validation set. We then performed binary classification on each pixel (poop-vs-background) using a threshold. Next, we rasterized the truth polygons. The corresponding truth and predicted pixels were accumulated into a confusion matrix, allowing us to compute standard metrics such as precision, recall, false positive rate, etc. [44] for the specific threshold. By sweeping a range of thresholds, we calculated the average precision (AP) and the area under the ROC curve (AUC). We computed all metrics using scikit-learn [43]. Due to the high number of true negative pixels, we preferred AP as the primary measure of model quality.

The details of the top model for each run, along with relevant hyperparameters, are presented in Table 4. This table also includes the results on the small, held out, test set for the top model.

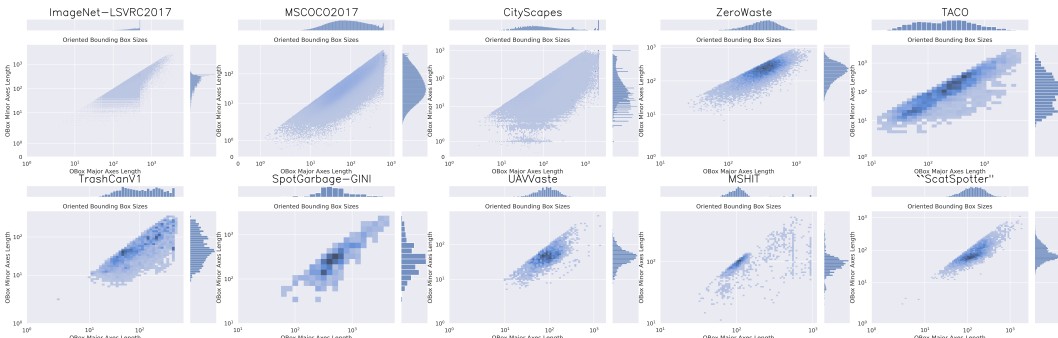

Figure 14: Oriented bounding box size distributions (log10 scale) of each dataset.

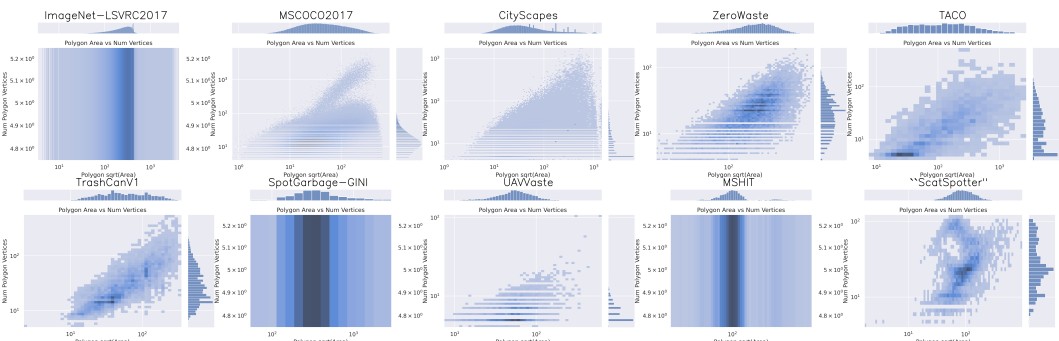

Figure 15: Polygon area versus number of vertices (log10 scale) for each dataset. The polygons with more vertices are more likely to be AI generated.

The results show strong performance on the validation set, with a maximum AP of $0.78$. However, while the test AP for this model is good, it is significantly lower at $0.51$. To investigate this discrepancy, we turned to qualitative analysis.

Qualitative results for the test, validation, and training sets are presented in Fig. 17. These examples illustrate both success and failure cases. The test and validation sets show clear responses to objects of interest, but the test set contains images of close-up and partially deteriorated poops. This suggests a bias in the dataset towards "fresh" poops taken from some distance.

Notably, the much larger training set also contains errors, indicating more information can be extracted from this dataset using hard-negative mining. There are clear difficult cases caused by sticks, leafs, pine cones, and dark areas on snow. We note that while compiling these results, we checked over 1000 images and discovered 14 cases where an object failed to be annotated, and it is likely that more are missed, but we believe these cases are rare.

Although focal loss was used, the current learning curriculum is likely under-weighting smaller distant objects. Our pixelwise evaluation metric is biased against this, which is a current limitation of our approach. Future work evaluating this dataset on an object-detection level can remedy this.

In Table 4 we only presented the top results. Here we've plotted the AP and AUC on the validation set for the top 5 AP-maximizing results from each of the 7 training runs. We also created a box-and-whisker plot for these top 5 results, which serves to assign a color and label to each training run. These plots are shown in Figure 18.

### C.1.1 Resource Usage

All models were trained on a single machine with an 11900k CPU and a 3090 GPU. At predict time, using one background worker, our models processed $416 \times 416$ patches at a rate of 20.93Hz with 94% GPU utilization.

To better understand the energy requirements of our model, particularly for potential deployment on mobile devices, we used CodeCarbon [30] to measure the resource usage during prediction and

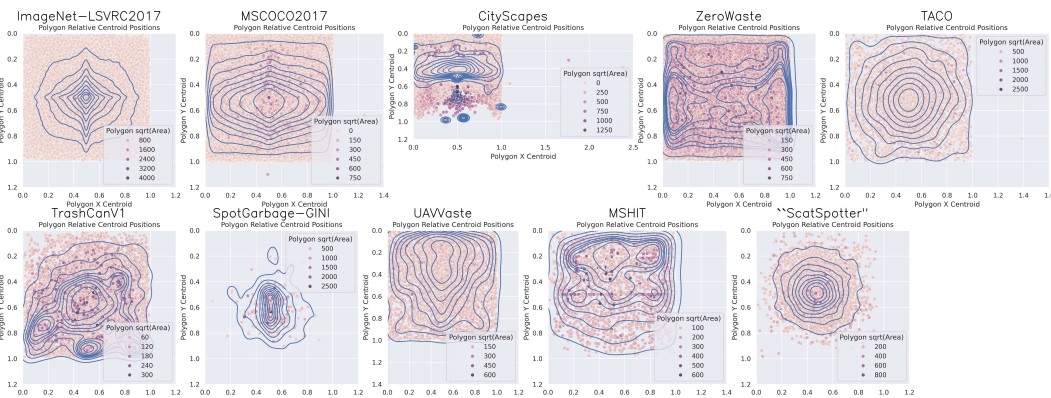

Figure 16: Polygon centroid relative distribution for each dataset. It is interesting to note patterns in this data. For instance, the outline of a street can be seen in CityScapes. In Zero Waste you can see the conveyor belt. ImageNet is more uniform. Ours is Gaussian distributed.

Table 4: Results for the best-performing models on the validation set across 7 hyperparameter configurations. The table provides detailed information about each configuration, including: 1) Configuration name (first column): a unique code identifying each training run used in the score scatter and box plots. 2) Varied hyperparameters (next three columns): specific values for learning rate, weight decay, and perturb scale that were used in each run. 3) Validation set performance (AP and AUC scores): metrics evaluating the model's performance on the validation set. 4) Test set performance (AP and AUC scores): metrics evaluating the model's performance on the test set using the same validation-maximizing models. Note that the top AP score over all models on the test set was 0.65, but it did not correspond to one of these validation runs used for model selection. Qualitative examples illustrating the performance of the top-scoring validation model listed here are provided in Fig. 17.

| config name | lr | weight_decay | perterb_scale | Validation (n=691) | | Test (n=30) | |
| | | | | AP | AUC | AP | AUC |
|---|---|---|---|---|---|---|---|
| D05 | 1e-4 | 1e-6 | 3e-6 | **0.7802** | **0.9943** | 0.5051 | 0.9125 |
| D03 | 1e-4 | 1e-5 | 3e-7 | 0.7758 | 0.9707 | 0.4346 | 0.8576 |
| D04 | 1e-4 | 1e-7 | 3e-7 | 0.7725 | 0.9818 | 0.4652 | 0.7965 |
| D02 | 1e-4 | 1e-6 | 3e-7 | 0.7621 | 0.9893 | **0.5167** | **0.9252** |
| D00 | 3e-4 | 3e-6 | 9e-7 | 0.7571 | 0.9737 | 0.4210 | 0.7766 |
| D01 | 1e-3 | 1e-5 | 3e-6 | 0.7070 | 0.9913 | 0.4607 | 0.9062 |
| D06 | 1e-4 | 1e-6 | 3e-8 | 0.6800 | 0.9773 | 0.4137 | 0.8157 |

evaluation. This analysis not only informs practical considerations but also helps us assess our contribution to the growing carbon footprint of AI [28]. The results for the 7 presented training experiments and the total 42 training experiments are reported in Table 5.

Direct measurement of resource usage during training is still under development, but we estimate the duration of each training run using indirect methods. We approximate energy consumption by assuming a constant power draw of 345W from the 3090 GPU during training. Emissions are estimated using a conversion ratio of 0.21 $\frac{\text{kgCO}_2}{\text{kWh}}$.

Based on the validation set's 691 images, we estimate that predicting on a single image on our desktop requires approximately 1.15 seconds and 0.13 Wh of energy. For context, typical mobile phones have a battery capacity of around 10 Wh and significantly less compute power than our desktop setup. While our models demonstrate the feasibility of training a strong detector from our dataset, they are not optimized for the mobile setting. To deploy our model on mobile devices, we will need to improve its efficiency or explore more efficient architectures.

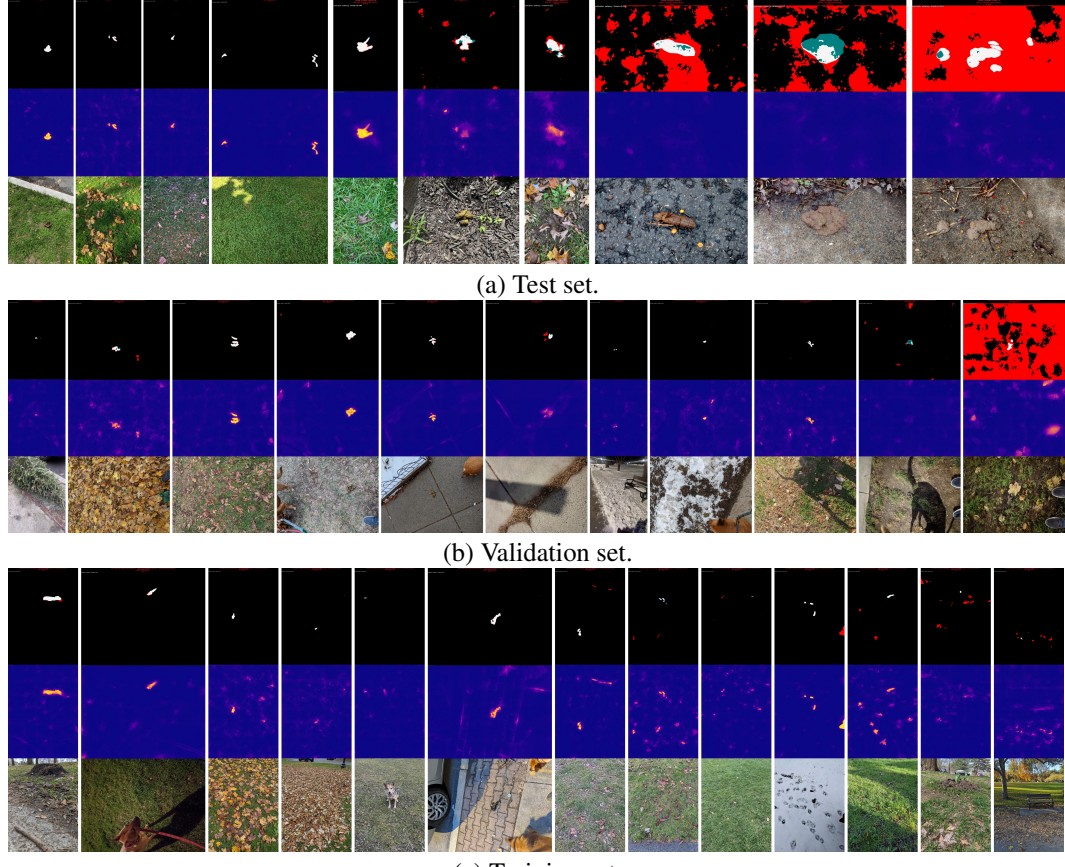

(a) Test set.

(b) Validation set.

(c) Training set.

Figure 17: Qualitative results using the top-performing model on the validation set, applied to a selection of images from the (a) test, (b) validation, and (c) training sets. Success cases are presented on the left, with failure cases increasing towards the right. Each figure is organized into three rows: Top row: Binarized classification map, where true positive pixels are shown in white, false positives in red, false negatives in teal, and true negatives in black. The threshold for binarization was chosen to maximize the F1 score for each image, showcasing the best possible classification of the heatmap. Middle row: The predicted heatmap, illustrating the model's output before binarization. Bottom row: The input image, providing context for the prediction. The majority of images in the test set exhibit qualitatively good results. Failure cases tend to occur with close-up images of older, sometimes partially deteriorated poops. These examples were manually selected and ordered to demonstrate dataset diversity in addition to representative results.

### C.1.2 Dataset Versions

There are two main versions of the dataset used in this paper. We can specify these using content-based identifiers. The version from 2024-07-03 has a IPFS CID of: `bafybeiedwp2zvmdyb2c2a` `xrcl455xfbv2mgdbhgkc3dile4dftiimwth2y` and a BitTorrent magnet of: `magnet:?xt=urn:` `btih:ee8d2c87a39ea9bfe48bef7eb4ca12eb68852c49`. The version from 2025-04-20 has an IPFS CID of: `bafybeia2uv3ea3aoz27ytiwbyudrjzblfuen47hm6tyfrjt6dgf6iadta4` and a BitTorrent magnet of: `magnet:?xt=urn:btih:27a2512ae93298f75544be6d2d629dfb186f86` `cf`. Note: the hash suffix of the magnet URL can be searched on `academictorrents.com`.

At the time of writing, the version of the dataset on HuggingFace is the latest, and we use git tags that correspond with the date of release and the IPFS CID to help identify dataset versions. However, unlike the decentralized methods, these are guaranteed to point to the expected version of the dataset. At the time of writing the HuggingFace URL is: `https://` `huggingface.co/datasets/[redactedforpeerreview]/scatspotter` and the Girder URL

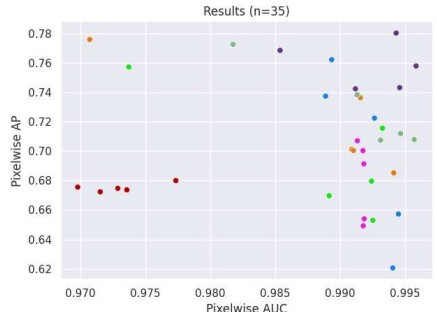

(a) AP and AUC of 35 checkpoints.

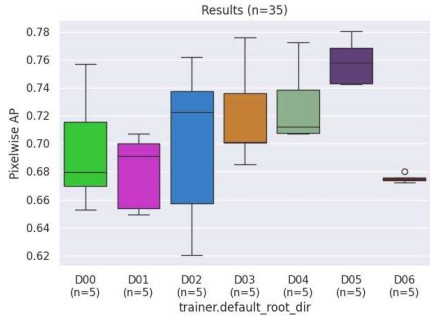

(b) AP of 35 checkpoints.

Figure 18: (a) Scatterplot of pixelwise average precision (AP) and Area Under the ROC curve (AUC) for the top 5 checkpoints on the validation set. Points of the same color represent checkpoints from the same training run, which used identical hyperparameters. (b) Box-and-whisker plot the AP values across the top 5 checkpoints evaluated on the validation set. For each run, corresponding varied hyperparameters and maximum APs are given in Table 4.

Table 5: Resources used for training, prediction, and evaluation. The "node" column is the pipeline stage: "train" for training, "pred" for heatmap prediction, and "eval" for pixelwise heatmap evaluation. The "resource" column lists the resource type: time, energy, or emissions. The "total" and "$\mu$" columns show the total and average consumptions, and the "n" column indicates the frequency of each stage (e.g., across different hyperparameters). Train rows marked with an asterisk (*) are based on indirect measurements.

(a) Presented experiment resources.

| Node | Resource | Total | $\mu$ | n |
|---|---|---|---|---|
| eval | time | 14.24 hours | 0.41 hours | 35 |
| pred | time | 11.97 hours | 0.34 hours | 35 |
| pred | energy | 8.76 kWh | 0.25 kWh | 35 |
| pred | emissions | 1.84 $CO_2$kg | 0.05 $CO_2$kg | 35 |
| train* | time | 39.22 days | 5.60 days | 7 |
| train* | energy | 324.75 kWh | 46.39 kWh | 7 |
| train* | emissions | 68.20 $CO_2$kg | 9.74 $CO_2$kg | 7 |

(b) All experiment resources.

| Node | Resource | Total | $\mu$ | n |
|---|---|---|---|---|
| eval | time | 5.84 days | 0.35 hours | 399 |
| pred | time | 7.29 days | 0.44 hours | 399 |
| pred | energy | 102.83 kWh | 0.26 kWh | 399 |
| pred | emissions | 21.6 $CO_2$kg | 0.05 $CO_2$kg | 399 |
| train* | time | 158.95 days | 3.78 days | 42 |
| train* | energy | 1,316.07 kWh | 31.34 kWh | 42 |
| train* | emissions | 276.37 $CO_2$kg | 6.58 $CO_2$kg | 42 |

512  is: https://data.[redactedforpeerreview].com/?#user/598a19658d777f7d33e9c18b/
513  folder/66b6bc7ef87a980650f41f98.

