# OpenReview forum: "``ScatSpotter'' --- A Dog Poop Detection Dataset"
_NeurIPS.cc/2025/Datasets_and_Benchmarks_Track — Submitted to NeurIPS 2025 Datasets and Benchmarks Track_

### Official Review · Reviewer_LGt7 · 2025-06-09

**Rating:** 4
**Confidence:** 5

**Summary:**

This paper presents a dataset comprising annotations of 9,000 dog feces instances. The author details the data collection and annotation procedures and further validates the dataset by training a segmentation model based on Mask R-CNN with a ViT backbone. In addition, the paper includes analyses related to clustering and attribute characterization.

**Additional Feedback:**

Indeed, I find this to be an interesting piece of work. If the authors can approach it with a more research-oriented mindset and treat it in the same rigorous manner as existing segmentation and detection benchmarks, I believe the paper still holds strong potential.

However. Current experiments are somewhat insufficient to be fully released as a benchmark. While the information density and overall contribution are still some distance away from the standards typically seen in NeurIPS-accepted papers, I believe there is potential. If the authors can provide more comprehensive experiments in the rebuttal, I would be willing to raise my score.

**Dataset Code Accessibility:**

Yes

**Dataset Code Comments:**

The dataset is well annotated on the Huggingface platform.

**Ethical Comments:**

The dataset mentioned in this paper only contains dog feces. No Ethics impact human beings or commercial policies.

**Ethical Considerations:**

No, there are no or only very minor ethics concerns

**Final Justification:**

**First, in response to AC’s concern:** I believe this paper, overall, presents an dog feces image dataset, a category for which there was previously a lack of publicly available benchmarks. Therefore, I feel the paper meets the standards of the DB track. It also holds value for applications such as robotic vacuum cleaners.

My main concern was addressed by the authors in the rebuttal. While I acknowledge the importance of eco-friendliness, an academic paper must ensure that the main text is sufficiently detailed and self-contained. This issue has been appropriately resolved by the additional experimental results provided by the authors. They included new models and more commonly used segmentation metrics.

Overall, I believe the paper has reached the level of completeness required. I am willing to raise my score from 3 to 4. Furthermore, if there were a baseline model specifically designed for this task, it would meet the standard I expect for a score of 5.

**Limitations Weaknesses:**

However, this paper has several issues that should be addressed:

1. The analysis of the dataset hosting platform's efficiency(section 5), along with the discussion on carbon emissions and power consumption(from line 157 to line 165), feels somewhat like filler content. These aspects are not essential for an academic conference paper like NeurIPS and would be more appropriate in the appendix or the GitHub README. The remaining space would be better used to strengthen the paper with more substantial analyses, such as the ones I suggest below.

2. As a benchmark, the inclusion of only two models is somewhat limited. It would be beneficial to incorporate well-established segmentation backbones such as SegFormer, Swin Transformer, or ConvNeXt. The authors could also consider designing a model specifically tailored for feces detection—optimized for sparse, small-object detection—which could serve as a meaningful contribution in itself.

3. Given that this is a small-object detection task, there is a significant imbalance between black and white pixels for the ground truth masks. Evaluating model performance using metrics like F1 score or IoU would provide more informative and meaningful insights.

**Strengths Contributions:**

This work indeed fills a gap in the field of animal feces detection and segmentation. The dataset is sufficiently large, high-resolution, and the masks are manually annotated. From what I can tell, the task appears challenging and valuable enough, viewed from Hugging Face. The authors also mention potential applications such as developing autonomous cleaning robots or assisting animal owners in locating feces hidden in grass—both of which are reasonable use cases.

Author also provide detailed analysis about how the dataset is collected.

---

> ### Author Rebuttal · Authors · 2025-07-30
>
> Thank you for your interest in our work.  We appreciate your acknowledgment of the dataset's value and offer clarifications and expanded experiments in response to your suggestions.
>
> ### On Power Consumption and Hosting Study
>
> While we understand that Section 5 and our environmental impact analysis may appear tangential, we argue they are relevant to long-term reproducibility (via automated and verifiable experiments) and ethical rigor of AI research.  Nonetheless, we will streamline these sections and move extended content to the appendix to ensure the main paper prioritizes expanded model contributions.
>
> ### On a domain-optimized model
>
> While exploring a model tailored for this specific sub-domain of object detection would be interesting, that is beyond the scope of this work which has the primary goal of introducing a new dataset.  We agree that future work could explore a model tailored for amorphous, camouflaged, sparse small-object detection, and believe our dataset can serve as a strong foundation for such efforts.
>
>
> ### On Metric Expansion
>
> Thank you for the suggestions.
>
> **In response to the reviewer's request, we now report F1** and TPR (recall) for both bounding-box and pixel-level evaluations using operating points selected to maximize F1.  Note, IoU can be derived as `IoU = F1 / (2 - F1)`, and PPV (precision) as `PPV = (F1 * TPR) / (2 * TPR - F1)`.  The reported TPR and F1 provide a more interpretable result while maintaining the holistic AP evaluation.
>
> **In response to the reviewer's request for a more rigorous benchmark, we expanded our evaluation to include two recent architectures**: GroundingDINO (which uses a Swin Transformer backbone) and YOLO-v9.  GroundingDINO was tested both in zero-shot (using 10 prompt variants) and fine-tuned settings.  Ablated zero-shot prompt results are available in the response to Reviewer 1 (TVgg03).
>
> GroundingDINO poorly in zero-shot settings (test/validation AP of 0.22/0.07), suggesting that this domain is not represented in existing foundational models.  However, when fine-tuned on our dataset, achieved a new state-of-the-art performance of 0.70 (test) / 0.69 (validation).  This establishes a strong upper baseline and demonstrates the dataset's practical value for driving progress in small-object detection, while highlighting that challenges remain.
>
> Revised and expanded results are as follows:
>
> ---
>
> **Main Test Box & Pixel Results (n=121)**
>
> | Model                | AP-box    | AUC-box   | F1-box    | TPR-box   | AP-pixel  | AUC-pixel | F1-pixel  | TPR-pixel |
> |----------------------|-----------|-----------|-----------|-----------|-----------|-----------|-----------|-----------|
> | MaskRCNN-pretrained  | 0.613     | **0.698** | 0.650     | 0.596     | **0.811** | **0.849** | **0.779** | **0.732** |
> | MaskRCNN-scratch     | 0.254     | 0.465     | 0.345     | 0.300     | 0.385     | 0.798     | 0.408     | 0.439     |
> | VIT-sseg-scratch     | 0.393     | 0.404     | 0.517     | 0.408     | 0.407     | 0.819     | 0.479     | 0.370     |
> | GroundingDINO-tuned  | **0.700** | 0.666     | **0.758** | **0.682** | ---       | ---       | ---       | ---       |
> | GroundingDINO-zero   | 0.228     | 0.303     | 0.393     | 0.377     | ---       | ---       | ---       | ---       |
> | YOLOv9-pretrained    | 0.441     | 0.551     | 0.507     | 0.498     | ---       | ---       | ---       | ---       |
> | YOLOv9-scratch       | 0.362     | 0.358     | 0.480     | 0.372     | ---       | ---       | ---       | ---       |
>
> ---
>
> **Main Validation Box & Pixel Results (n=691)**
>
> | Model                | AP-box    | AUC-box   | F1-box    | TPR-box   | AP-pixel  | AUC-pixel | F1-pixel  | TPR-pixel |
> |----------------------|-----------|-----------|-----------|-----------|-----------|-----------|-----------|-----------|
> | MaskRCNN-pretrained  | 0.612     | **0.721** | 0.625     | 0.573     | **0.744** | 0.906     | **0.738** | 0.676     |
> | MaskRCNN-scratch     | 0.255     | 0.576     | 0.349     | 0.311     | 0.434     | 0.891     | 0.482     | 0.501     |
> | VIT-sseg-scratch     | 0.476     | 0.532     | 0.596     | 0.510     | 0.757     | **0.974** | 0.736     | **0.695** |
> | GroundingDINO-tuned  | **0.691** | 0.631     | **0.743** | **0.684** | ---       | ---       | ---       | ---       |
> | GroundingDINO-zero   | 0.078     | 0.210     | 0.197     | 0.252     | ---       | ---       | ---       | ---       |
> | YOLOv9-pretrained    | 0.411     | 0.595     | 0.496     | 0.424     | ---       | ---       | ---       | ---       |
> | YOLOv9-scratch       | 0.331     | 0.409     | 0.443     | 0.367     | ---       | ---       | ---       | ---       |
>
> ---
>
> *Note: We identified that results in the submitted paper were inadvertently reported on an older 30-image test set (d8988f8c). All results have been updated to the correct 121-image test set (6cb3b6ff); conclusions remain unchanged.*
>
> Further details are available in the response to Reviewer 1 (TVgg03).
>
>
> We hope these expanded benchmarks and metrics demonstrate the rigor the reviewer called for.  We believe the revised submission now meets the standards for a strong paper, and would greatly appreciate reconsideration of the score in light of these additions.

---

> > ### Comment · Reviewer_LGt7 · 2025-08-05
> >
> > My main concern was addressed by the authors in the rebuttal. While I acknowledge the importance of eco-friendliness, an academic paper must ensure that the main text is sufficiently detailed and self-contained. This issue has been appropriately resolved by the additional experimental results provided by the authors. They included new models and more commonly used segmentation metrics.
> >
> > Overall, I believe the paper has reached the level of completeness required. I am willing to raise my score from 3 to 4. Furthermore, if there were a baseline model specifically designed for this task, it would meet the standard I expect for a score of 5.

---

> ### Author Response · Authors · 2025-08-05
>
> Thank you for your review and pushing us towards higher scientific standards. The next phase of our work will be exploring a wider range of mobile architectures and deployment of the system in a free mobile app. We will consider your suggestion and think about novel modifications of these networks that better tailor them towards the task.

---

### Official Review · Reviewer_NeGf · 2025-06-30

**Rating:** 5
**Confidence:** 3

**Summary:**

This paper introduces a novel dataset for dog waste detection, featuring high-quality annotations and a unique "before/after/negative" collection protocol. Its exploration of decentralized data distribution (IPFS, BitTorrent) is particularly forward-thinking. While the dataset is robust, its geographic bias and small test set could limit generalizability. Recommended for acceptance with minor revisions to expand test data and model benchmarks.

**Dataset Code Accessibility:**

Yes

**Ethical Considerations:**

No, there are no or only very minor ethics concerns

**Final Justification:**

I appreciate the authors' thoughtful response and clarifications provided regarding the dataset limitations and distribution, as well as the additional model results.

On Dataset Limitations:
I understand and acknowledge the authors' explanation regarding the geographic and subject biases in the dataset. It is reassuring to know that intentional efforts were made to include a diverse range of settings, and that 40% of the annotated images came from unowned animals. The authors' strategy to design a test set independent of the author-collected images helps mitigate bias, which is a commendable approach.

On the Expanded Results:
The revised and expanded results that include additional metrics (F1 and TPR for bounding-box and pixel-level evaluations) significantly improve the comprehensiveness of the paper. The updated results, along with the correction of the test set, strengthen the reliability of the findings. The authors' efforts in providing a more detailed and accurate evaluation enhance the overall credibility of the work.

I believe these clarifications, especially regarding the dataset, the reproducibility mechanisms, and the updated results, address the concerns raised in the review.

**Limitations Weaknesses:**

The primary limitation is the small independent test set (121 images), which may not fully capture model generalization. Geographic bias (data from a single city) could also affect real-world applicability. Future work could expand the dataset’s diversity and explore lightweight architectures (e.g., YOLO) for mobile deployment. The peer-discovery issues in decentralized distribution warrant further investigation under varied network conditions.

**Strengths Contributions:**

This paper introduces a novel and practical dataset for dog waste detection, with potential applications in urban cleanliness and smart devices. The "before/after/negative" (BAN) protocol enhances data diversity, while the inclusion of 9,000+ images with 6,000+ polygon annotations ensures robustness. The comparison of centralized (HuggingFace, Girder) and decentralized (IPFS, BitTorrent) distribution methods is particularly insightful, addressing long-term accessibility challenges. Baseline models (ViT, Mask R-CNN) provide a solid reference, and the analysis of computational costs aligns with sustainability concerns in AI research.

---

> ### Author Rebuttal · Authors · 2025-07-30
>
> Thank you for your constructive review, appreciation of our focus on dataset distribution, and recognition of our efforts to quantify our environmental impact.
>
> We are grateful for the reviewer's positive impression of our work as well as their appreciation of our focus on dataset distribution, which we believe is a novel aspect of our work. While our environmental impact is small, we feel ethically compelled to quantify it and we thank the reviewer for the recognition of these efforts.
>
> ### On Dataset/Test Limitations
>
> The small independent test set stems from an early design choice to exclude all author-captured images. While this limits test diversity, it ensures reduced collection bias. We make this limitation very clear and compensate by providing results on the larger validation set.
>
> ###  On Distribution Study
>
> Our decentralized benchmarking was intentionally carried out in an uncontrolled setting as an observational study of how real dataset performs under real world network conditions.  Our github repo contains logs from all trials as well as programmatic instructions to expand these statistics with new observations.  This provides a basis for other researchers with different network conditions to further investigate the issue.
>
> ### On Model Scope
>
> We agree the original benchmark was too narrow.  **We now include results for YOLOv9** and GroundingDINO, with fine-tuned GroundingDINO achieving a new best performance (0.70/0.69 on the test/validation set).  These results reinforce the utility of our dataset and the challenge it poses even to state-of-the-art detectors.
>
> We expand all reported metrics to include F1 and TPR (recall) for both bounding-box and pixel-level evaluations using operating points selected to maximize F1.
> Revised and expanded results are as follows:
>
> ---
>
> **Main Test Box & Pixel Results (n=121)**
>
> | Model                | AP-box    | AUC-box   | F1-box    | TPR-box   | AP-pixel  | AUC-pixel | F1-pixel  | TPR-pixel |
> |----------------------|-----------|-----------|-----------|-----------|-----------|-----------|-----------|-----------|
> | MaskRCNN-pretrained  | 0.613     | **0.698** | 0.650     | 0.596     | **0.811** | **0.849** | **0.779** | **0.732** |
> | MaskRCNN-scratch     | 0.254     | 0.465     | 0.345     | 0.300     | 0.385     | 0.798     | 0.408     | 0.439     |
> | VIT-sseg-scratch     | 0.393     | 0.404     | 0.517     | 0.408     | 0.407     | 0.819     | 0.479     | 0.370     |
> | GroundingDINO-tuned  | **0.700** | 0.666     | **0.758** | **0.682** | ---       | ---       | ---       | ---       |
> | GroundingDINO-zero   | 0.228     | 0.303     | 0.393     | 0.377     | ---       | ---       | ---       | ---       |
> | YOLOv9-pretrained    | 0.441     | 0.551     | 0.507     | 0.498     | ---       | ---       | ---       | ---       |
> | YOLOv9-scratch       | 0.362     | 0.358     | 0.480     | 0.372     | ---       | ---       | ---       | ---       |
>
> ---
>
> **Main Validation Box & Pixel Results (n=691)**
>
> | Model                | AP-box    | AUC-box   | F1-box    | TPR-box   | AP-pixel  | AUC-pixel | F1-pixel  | TPR-pixel |
> |----------------------|-----------|-----------|-----------|-----------|-----------|-----------|-----------|-----------|
> | MaskRCNN-pretrained  | 0.612     | **0.721** | 0.625     | 0.573     | **0.744** | 0.906     | **0.738** | 0.676     |
> | MaskRCNN-scratch     | 0.255     | 0.576     | 0.349     | 0.311     | 0.434     | 0.891     | 0.482     | 0.501     |
> | VIT-sseg-scratch     | 0.476     | 0.532     | 0.596     | 0.510     | 0.757     | **0.974** | 0.736     | **0.695** |
> | GroundingDINO-tuned  | **0.691** | 0.631     | **0.743** | **0.684** | ---       | ---       | ---       | ---       |
> | GroundingDINO-zero   | 0.078     | 0.210     | 0.197     | 0.252     | ---       | ---       | ---       | ---       |
> | YOLOv9-pretrained    | 0.411     | 0.595     | 0.496     | 0.424     | ---       | ---       | ---       | ---       |
> | YOLOv9-scratch       | 0.331     | 0.409     | 0.443     | 0.367     | ---       | ---       | ---       | ---       |
>
> ---
>
> *Note: We identified that results in the submitted paper were inadvertently reported on an older 30-image test set (d8988f8c). All results have been updated to the correct 121-image test set (6cb3b6ff); conclusions remain unchanged.*
>
> Further details are available in the response to Reviewer 1 (TVgg03).
>
> We hope these updates reinforce the reviewer's positive assessment and provide the clarity needed to raise confidence in the submission.

---

> > ### Comment · Reviewer_NeGf · 2025-08-09
> >
> > Thank you for authors' response. Multiple models were used to validate the dataset, which addressed my concern, I have no further suggestions. I want to raise my original score to 5.

---

### Official Review · Reviewer_Ux3F · 2025-07-02

**Rating:** 5
**Confidence:** 3

**Summary:**

The paper introduces a dataset of images of dog poop, which poses various challenges (small, low contrast to environment, highly variable in appearance) for image processing algorithms. The dataset contains over 9000 images with 6000 annotated instances, and for many scenes has three images that depict the same region *before* the dog went to work, *after* the dog went to work, and another close by region without any dog action (*negative*). Two standard detector models are trained on the dataset to validate its difficulty.

Additionally, the paper explores technical ways of distributing the dataset, with a focus on availability and speed.

**Dataset Code Accessibility:**

Yes

**Dataset Code Comments:**

Both dataset and evaluation code are available on standard platforms.

**Ethical Comments:**

Reviewer does not see any ethical concerns w.r.t. images of dog poop.

**Ethical Considerations:**

No, there are no or only very minor ethics concerns

**Final Justification:**

The original strengths and weaknesses still stand. The rebuttal is appreciated and the additions to the paper are valuable. The paper has the potential to steer interest into the specific direction of "dog poop detection in the wild", more general into small-scale outdoor waste and object detection.

**Limitations Weaknesses:**

* Not really a weakness, and certainly up to discussion, but the discussion about dataset distribution (Chapter 5) is tangential to the main contribution of the paper. While reproducibility of methods is an issue, large datasets are routinely distributed robustly in the computer vision community. Having larger datasets disappear happens rarely. The paper would be just as strong without Chapter 5.

* The dataset is somewhat limited in scope; most images were taken at geographically similar locations, most are from the same three dogs, probably with a somewhat constant and similar diet, and taken by the same person which probably introduces a bias regarding camera placement and settings.

**Strengths Contributions:**

* Novelty, potentially high impact: There is no public large-scale dataset regarding dog poop yet. That, in combination with uniquely challenging aspects regarding target size and visual aspects, can lead to a high impact of this paper, opening a path for both method development and additional datasets.

* Practical interest: The topic can be of practical interest, with the rise of robots that automatically clean public places from litter and - potentially - dog poop.

* Well though-off setup: The somewhat unusual before/after/negative setup is well thought off and allows for training protocols that would be unavailable otherwise.

* The paper is very well written, easy to follow, everything is well motivated. The dataset is analyzed in detail and fairly compared to other datasets, giving a very good intuition about its strengths, challenges and potential future improvements. The authors also propose a standard train/validation/test-split, which is appreciated.

---

> ### Author Rebuttal · Authors · 2025-07-30
>
> We thank the reviewer for their support and appreciation of our work.
>
> ### On Dataset Limitations
>
> Geographic and subject biases are an acknowledged limitation.
> While most images were captured by the authors, the dataset includes diverse settings (seasonal, lighting,
>   weather) and both known and unknown dogs.
> We intentionally designed the test set to be independent of author imagery in order to mitigate bias from
>   author-collected images.
>
> While we do not have a proper measurement, we estimate approximately 40% of annotated images are from
>   unowned animals encountered during collection.
>
> ### On Dataset Distribution
>
> While we agree Section 5 may be tangential to some readers, content addressable data is important for
>   automated and verifiable reproduction of experiments.
> It is true that it is rare for large datasets to disappear, but the risk remains.
> Coupling the mechanism for download and verification adds a layer of protection missing in most standard
>   datasets.
> That said, we will move less essential benchmarking details to the appendix to make room for an expanded set
>   of model results in the main text.
>
> We expand all reported metrics to include F1 and TPR (recall) for both bounding-box and pixel-level evaluations using operating points selected to maximize F1.
> Revised and expanded results are as follows:
>
> ---
>
> **Main Test Box & Pixel Results (n=121)**
>
> | Model                | AP-box    | AUC-box   | F1-box    | TPR-box   | AP-pixel  | AUC-pixel | F1-pixel  | TPR-pixel |
> |----------------------|-----------|-----------|-----------|-----------|-----------|-----------|-----------|-----------|
> | MaskRCNN-pretrained  | 0.613     | **0.698** | 0.650     | 0.596     | **0.811** | **0.849** | **0.779** | **0.732** |
> | MaskRCNN-scratch     | 0.254     | 0.465     | 0.345     | 0.300     | 0.385     | 0.798     | 0.408     | 0.439     |
> | VIT-sseg-scratch     | 0.393     | 0.404     | 0.517     | 0.408     | 0.407     | 0.819     | 0.479     | 0.370     |
> | GroundingDINO-tuned  | **0.700** | 0.666     | **0.758** | **0.682** | ---       | ---       | ---       | ---       |
> | GroundingDINO-zero   | 0.228     | 0.303     | 0.393     | 0.377     | ---       | ---       | ---       | ---       |
> | YOLOv9-pretrained    | 0.441     | 0.551     | 0.507     | 0.498     | ---       | ---       | ---       | ---       |
> | YOLOv9-scratch       | 0.362     | 0.358     | 0.480     | 0.372     | ---       | ---       | ---       | ---       |
>
> ---
>
> **Main Validation Box & Pixel Results (n=691)**
>
> | Model                | AP-box    | AUC-box   | F1-box    | TPR-box   | AP-pixel  | AUC-pixel | F1-pixel  | TPR-pixel |
> |----------------------|-----------|-----------|-----------|-----------|-----------|-----------|-----------|-----------|
> | MaskRCNN-pretrained  | 0.612     | **0.721** | 0.625     | 0.573     | **0.744** | 0.906     | **0.738** | 0.676     |
> | MaskRCNN-scratch     | 0.255     | 0.576     | 0.349     | 0.311     | 0.434     | 0.891     | 0.482     | 0.501     |
> | VIT-sseg-scratch     | 0.476     | 0.532     | 0.596     | 0.510     | 0.757     | **0.974** | 0.736     | **0.695** |
> | GroundingDINO-tuned  | **0.691** | 0.631     | **0.743** | **0.684** | ---       | ---       | ---       | ---       |
> | GroundingDINO-zero   | 0.078     | 0.210     | 0.197     | 0.252     | ---       | ---       | ---       | ---       |
> | YOLOv9-pretrained    | 0.411     | 0.595     | 0.496     | 0.424     | ---       | ---       | ---       | ---       |
> | YOLOv9-scratch       | 0.331     | 0.409     | 0.443     | 0.367     | ---       | ---       | ---       | ---       |
>
> ---
>
> *Note: We identified that results in the submitted paper were inadvertently reported on an older 30-image test set (d8988f8c). All results have been updated to the correct 121-image test set (6cb3b6ff); conclusions remain unchanged.*
>
> Further details are available in the response to Reviewer 1 (TVgg03).
>
> We hope that these clarifications increase the reviewer's confidence in their positive assessment of our work.

---

### Official Review · Reviewer_TVgg · 2025-07-03

**Rating:** 3
**Confidence:** 4

**Summary:**

This paper presents “ScatSpotter”, a high-resolution dataset of 9,000+ images with polygon annotations for dog feces detection, along with baseline results using MaskRCNN and Vision Transformer models.

**Dataset Code Accessibility:**

Yes

**Dataset Code Comments:**

The paper releases code and uses open-source baselines for experiments, which strengthens reproducibility.

**Ethical Considerations:**

No, there are no or only very minor ethics concerns

**Final Justification:**

While the paper demonstrates unique value and interesting potential, and the rebuttal is appreciated, I believe the overall completeness can be further strengthened.

**Limitations Weaknesses:**

- The dataset has geographic bias (primarily from a single city and a few dogs), which may limit generalizability.
- The independent test set is small (only 121 images), reducing the robustness of model evaluation.
- Consider evaluating a broader range of object detection and segmentation models (e.g., YOLO variants, MobileNet-based detectors, and recent transformer-based approaches) to provide a more comprehensive benchmark.
- The proposed solution appears somewhat outdated in the era of rapidly advancing multimodal large models. I suggest the authors include experiments using open-ended detectors such as Grounding DINO, and provide a comparative analysis between open-ended object detection (with few-shot training) and closed-ended object detection. Such an investigation would make the conclusions more compelling and highlight new insights.
- The paper lacks methodological novelty and insights in the context of object detection. In my view, “dog feces detection” should not be treated as a standalone problem but rather as a subtask of general object detection, with the primary distinction being the data source and distribution. I recommend that the authors clarify how their approach provides unique contributions or innovations compared to existing general object detection methods.

**Strengths Contributions:**

- Provides a high-resolution, well-annotated, and open dog feces detection dataset (ScatSpotter), filling a gap in this domain.
- Innovatively employs a “before/after/negative (BAN)” collection protocol to increase negative sample diversity and enhance model generalization.

---

> ### Author Rebuttal · Authors · 2025-07-30
>
> Thank you for your feedback.  We appreciate your recognition of our dataset's niche and the BAN collection protocol. We agree with your suggestion to open-ended object detection experiments.  Below we address your concerns and provide new experimental results on GroundingDINO and YOLO-v9 that strengthen the paper.
>
> ### On Geographic Bias and Test Size
>
> We acknowledge the geographic bias in our dataset.  However, the long collection period, environmental diversity (day/night, seasonal variation), and mix of known and unknown dogs introduce substantial variability (see Fig.  3 and Fig.  4a-b).  The test set was intentionally designed to exclude author-collected images to reduce capture bias.  While this limits its size, we compensate by reporting results on both the test and validation set.
>
> ### On Test Dataset Size
>
> The size of the test dataset is due to a design decision set at the start of the project.  We wanted to ensure that the test set contained no images captured by the authors.  Thus we relied on external contributions.  We take care to acknowledge this limitation, and report results on the larger validation set in an effort to compensate.
>
> ### On Novelty
>
> While feces detection is indeed a subdomain of object detection, it remains underexplored.  The novelty lies in the class itself, which presents challenges - amorphous shape, camouflage, small size in high resolution images - that, in combination, differ significantly from standard benchmarks.
>
> The uniqueness of our dataset is quantified in Table 1 and Figure 2 of our paper, where our dataset has the second largest image area to object area ratio (i.e. object sparsity) surpassed only by UAVVaste.  However, our dataset is considerably larger and collected in a ground-based setting, whereas UAVVaste involves aerial imagery, where such "needle-in-a-haystack" detection problems are more common.  Our dataset enables exploration of this niche, with potential applications in robotics and accessibility tools.  In our expanded results we show that foundational zero-shot models like GroundingDINO underperform substantially in this domain, reinforcing its academic and practical interest.
>
> ### On a Broader Range of Models
>
> We agree that evaluation across more architectures strengthens the benchmark.  **We have added results from GroundingDINO (172e6 params) (zero-shot and fine-tuned) and YOLOv9 (50e6) (pretrained and trained-from-scratch).**  In zero-shot settings, GroundingDINO performed poorly (AP 0.07-0.22) over 10 prompt variations, confirming the category is underrepresented.  Fine-tuning GroundingDINO achieved 0.70 AP (test), outperforming all prior models.  Lightweight YOLOv9 models performed well, but worse than other models, with pretrained AP of 0.44 (test).  We expand all reported metrics to include F1 and TPR (recall) for both bounding-box and pixel-level evaluations using operating points selected to maximize F1.
> Full results for the new prompt ablation and main results tables below:
>
> ---
>
> **Validation Prompt Ablation of Zero-Shot Grounding DINO (n=691)**
>
> | Prompt       |   AP        |   AUC       |   F1       |  TPR       |
> |--------------|-------------|-------------|------------|------------|
> | droppings    | 0.0233      | 0.0955      | 0.1386     | 0.2264     |
> | petwaste     | 0.0351      | 0.1429      | 0.1527     | 0.2456     |
> | poop         | 0.0380      | 0.1035      | 0.1651     | 0.1626     |
> | dogpoop      | 0.0470      | 0.1645      | 0.1688     | 0.2041     |
> | caninefeces  | 0.0511      | 0.1632      | 0.1817     | 0.2918     |
> | turd         | 0.0539      | 0.1754      | 0.1786     | 0.2185     |
> | feces        | 0.0647      | 0.2076      | 0.1796     | 0.2695     |
> | excrement    | 0.0709      | 0.2166      | 0.1969     | 0.2775     |
> | dogfeces     | 0.0748      | 0.2071      | 0.1977     | **0.3094** |
> | animalfeces  | **0.0779**  | **0.2103**  | **0.1973** | 0.2519     |
>
> ---
>
> **Test Prompt Ablation of Zero-Shot Grounding DINO (n=121)**
>
> | Prompt       |    AP      |   AUC      |   F1       |  TPR       |
> |--------------|------------|------------|------------|------------|
> | droppings    | 0.0764     | 0.1429     | 0.2718     | 0.3004     |
> | feces        | 0.1628     | 0.2631     | 0.3228     | 0.3901     |
> | caninefeces  | 0.1664     | 0.2420     | 0.3666     | 0.3946     |
> | poop         | 0.1727     | 0.1784     | 0.3123     | 0.2556     |
> | petwaste     | 0.1979     | 0.2484     | 0.3510     | 0.3408     |
> | animalfeces  | 0.2280     | 0.3031     | 0.3925     | 0.3766     |
> | dogfeces     | 0.2284     | 0.2921     | **0.3990** | 0.3811     |
> | dogpoop      | 0.2411     | 0.2757     | 0.3796     | 0.3856     |
> | excrement    | 0.2520     | 0.3138     | 0.3907     | **0.4170** |
> | turd         | **0.2712** | **0.3169** | 0.3940     | 0.3542     |
>
> ---
>
> *Note: Prompt ablation results will be included in the appendix. The "animalfeces" prompt, selected on the validation set, is used for main zero-shot results.*
>
>
> The new main results are as follows:
>
> ---
>
> **Main Test Box & Pixel Results (n=121)**
>
> | Model                | AP-box    | AUC-box   | F1-box    | TPR-box   | AP-pixel  | AUC-pixel | F1-pixel  | TPR-pixel |
> |----------------------|-----------|-----------|-----------|-----------|-----------|-----------|-----------|-----------|
> | MaskRCNN-pretrained  | 0.613     | **0.698** | 0.650     | 0.596     | **0.811** | **0.849** | **0.779** | **0.732** |
> | MaskRCNN-scratch     | 0.254     | 0.465     | 0.345     | 0.300     | 0.385     | 0.798     | 0.408     | 0.439     |
> | VIT-sseg-scratch     | 0.393     | 0.404     | 0.517     | 0.408     | 0.407     | 0.819     | 0.479     | 0.370     |
> | GroundingDINO-tuned  | **0.700** | 0.666     | **0.758** | **0.682** | ---       | ---       | ---       | ---       |
> | GroundingDINO-zero   | 0.228     | 0.303     | 0.393     | 0.377     | ---       | ---       | ---       | ---       |
> | YOLOv9-pretrained    | 0.441     | 0.551     | 0.507     | 0.498     | ---       | ---       | ---       | ---       |
> | YOLOv9-scratch       | 0.362     | 0.358     | 0.480     | 0.372     | ---       | ---       | ---       | ---       |
>
> ---
>
> **Main Validation Box & Pixel Results (n=691)**
>
> | Model                | AP-box    | AUC-box   | F1-box    | TPR-box   | AP-pixel  | AUC-pixel | F1-pixel  | TPR-pixel |
> |----------------------|-----------|-----------|-----------|-----------|-----------|-----------|-----------|-----------|
> | MaskRCNN-pretrained  | 0.612     | **0.721** | 0.625     | 0.573     | **0.744** | 0.906     | **0.738** | 0.676     |
> | MaskRCNN-scratch     | 0.255     | 0.576     | 0.349     | 0.311     | 0.434     | 0.891     | 0.482     | 0.501     |
> | VIT-sseg-scratch     | 0.476     | 0.532     | 0.596     | 0.510     | 0.757     | **0.974** | 0.736     | **0.695** |
> | GroundingDINO-tuned  | **0.691** | 0.631     | **0.743** | **0.684** | ---       | ---       | ---       | ---       |
> | GroundingDINO-zero   | 0.078     | 0.210     | 0.197     | 0.252     | ---       | ---       | ---       | ---       |
> | YOLOv9-pretrained    | 0.411     | 0.595     | 0.496     | 0.424     | ---       | ---       | ---       | ---       |
> | YOLOv9-scratch       | 0.331     | 0.409     | 0.443     | 0.367     | ---       | ---       | ---       | ---       |
>
> ---
>
> *Note: We identified that results in the submitted paper were inadvertently reported on an older 30-image test set (d8988f8c). All results have been updated to the correct 121-image test set (6cb3b6ff); conclusions remain unchanged.*
>
> We hope these additions clarify the primary contribution of our submission: a novel, high-resolution dataset focused on a uniquely difficult object category, with small, camouflaged, and amorphous instances annotated at the polygon level. We respectfully encourage the reviewer to reconsider their score in light of the expanded benchmarks and clarified framing of our contribution.

---

### Note · Authors · 2025-08-12

**Author Final Remarks - Summary for AC**

**Primary contribution:**

* **Novel, high-resolution, polygon-annotated dataset** for a uniquely difficult small-object category - dog feces - collected with a **before/after/negative (BAN) protocol** to increase negative sample diversity and generalization.
* Distinct from standard benchmarks (amorphous shape, camouflage, extreme sparsity) and **underrepresented in current foundational vision models**.
* Dataset, code, and splits are **openly available** and **hash-verifiable**.

**Main results:**

* Independent test set is small **by design** (no author-collected images) - transparent about limitation.
* Combined with larger validation set, results demonstrate convincing, high-quality detection that:

    * Enables practical applications **right now**.
    * Clearly leaves **headroom for future research** with our dataset.

**Improvements in response to reviews:**

* Added **GroundingDINO** and **YOLOv9** benchmarks, with notable results from GroundingDINO:

  * Zero-shot Box AP **0.23 (test) / 0.08 (val)** -> reveals **domain absence** in current high-quality datasets.
  * Fine-tuned Box AP **0.70 (test) / 0.69 (val)** -> dataset is **sufficiently large** to advance foundational models.
* Reported **F1 and TPR** for both bounding-box and pixel-level results.
* Streamlined tangential sections; retained novel **dataset distribution study**, highlighted by one reviewer for its forward-thinking nature.

**Reviewer context:**

* All reviewers identified similar weaknesses (model diversity, tangential sections, test set size), which we addressed concretely: added models/metrics and streamlined text; test-set size is acknowledged and mitigated by an author-independent design plus results on a larger validation set.
* R1:TVgg - no post-rebuttal response; concerns addressed & main points explicitly resolved.
* R2:Ux3F - initially positive, no concerns after rebuttal.
* R3:NeGf - engaged post-rebuttal; raised score from 4 -> 5.
* R4:LGt7 - engaged post-rebuttal; raised score from 3 -> 4.

**Final position:**

* Reviews began positive and trended upward, with at least two reviewers raising their scores.
* Paper now **meets D&B acceptance standards** - offers a **unique, open dataset**, rigorous baselines, multiple hosting backends, and **new insights into foundational model coverage**.

We thank all reviewers for their feedback, which has resulted in a significantly strengthened and focused paper.

---

### Decision · Program_Chairs · 2025-09-18

**Decision:**

Reject

**Comment:**

The authors introduce a new dataset for evaluating vision models' ability to detect and segment dog feces in images. The proposed dataset contains ~9000 images and ~6500 annotated instances, in the form of segmentation masks, of dog feces. While the reviewers identified many positives aspects of the dataset (e.g., the unique nature of the task), they also highlighted several concerns (see below). After reading the rebuttals and engaging in the discussion, the final recommendations were mixed, i.e., two reviewers recommended acceptance, but the other two were borderline.

Reviewer concerns:
* High performance of the evaluated models in the main paper, which calls into question how future proof the dataset is
* Performance only reported for a small number of older models in the main paper (although results for two newer models were presented in the rebuttal)
* Limited scope of the dataset, i.e., in essence it is a segmentation task containing relatively small object instances
* Geographical bias of the training data with a limited number of dogs and small test set (i.e., 121 images)

The reviewers felt that the rebuttal addressed some of their concerns, but not all of them. In light of the above, and given the lack of overwhelming support from the reviewers, this area chair is of the opinion that the paper is not yet ready for publication and thus recommends rejection.